# Gallium arsenide solar cells grown at rates exceeding 300 μm h$^{-1}$ by hydride vapor phase epitaxy

Wondwosen Metaferia [1], Kevin L. Schulte[1], John Simon[1], Steve Johnston [1] & Aaron J. Ptak [1]

We report gallium arsenide (GaAs) growth rates exceeding 300 μm h$^{-1}$ using dynamic hydride vapor phase epitaxy. We achieved these rates by maximizing the gallium to gallium monochloride conversion efficiency, and by utilizing a mass-transport-limited growth regime with fast kinetics. We also demonstrate gallium indium phosphide growth at rates exceeding 200 μm h$^{-1}$ using similar growth conditions. We grew GaAs solar cell devices by incorporating the high growth rate of GaAs and evaluated its material quality at these high rates. Solar cell growth rates ranged from 35 to 309 μm h$^{-1}$ with open circuit voltages ranging from 1.04 to 1.07 V. The best devices exceeded 25% efficiency under the AM1.5 G solar spectrum. The high open-circuit voltages indicate that high material quality can be maintained at these extremely high growth rates. These results have strong implications toward lowering the deposition cost of III-V materials potentially enabling the deposition of high efficiency devices in mere seconds.

[1] National Renewable Energy Laboratory, Golden, CO 80401, USA. Correspondence and requests for materials should be addressed to
W.M. (email: Wondwosen.metaferia@nrel.gov) or to A.J.P. (email: aaron.ptak@nrel.gov)

Single-junction GaAs solar cells exhibit record efficiencies of 29.1% and 30.5% under one-sun and concentrated illumination, respectively[1]. However, high material and manufacturing costs restrict III–V based solar cells to specialty applications, such as space power and high concentration systems, despite their many advantages over other solar technologies. It is therefore essential to reduce the cost of III–V epitaxial growth in order for these solar cells to reach larger markets. Different pathways to reduce the cost of III–V epitaxial growth, such as high-growth-rate metalorganic vapor phase epitaxy (MOVPE)[2–4] and close-spaced vapor transport (CSVT)[5], are actively being studied. Hydride vapor phase epitaxy (HVPE) is another alternative to current standard industrial processes that has promise for reducing the costs of III–V epitaxy[6,7]. HVPE replaces the expensive group III metalorganic precursors used in MOVPE with lower cost elemental sources, and offers the potential for higher $AsH_3$ utilization and significantly higher throughput. Furthermore, growth of highly uniform, large area (up to 8-inch diameter)[8], semiconductor films has been demonstrated by HVPE in an industrial setting.

HVPE in decades past had difficulties making the abrupt and low-defect heterointerfaces necessary for the creation of complex, high-efficiency device structures. The advent of dynamic-HVPE (D-HVPE)[9–11] has allowed HVPE to reemerge as a III–V growth technique that is now capable of producing III–V heterojunction devices, with demonstrated high-efficiency GaAs single-junction solar cells[6], GaInP/GaAs tandem solar cells[11], and tunnel junction interconnects[12]. These results validate D-HVPE's ability to deposit III–V alloys with low bulk defect densities and abrupt, defect-free interfaces, all without sacrificing throughput or quality. Our D-HVPE system contains two growth chambers, shown schematically in Fig. 1a, between which a substrate is translated to deposit multilayer structures. The general idea of substrate motion through different growth chambers is extendable to a fully in-line system allowing the formation of many complex structures, such as multijunction solar cells, distributed Bragg reflectors (DBRs) and heterojunction bipolar transistors (HBTs).

Part of the promise of HVPE lies in its ability to generate extremely high growth rates. Previously, a GaAs epilayer growth rate of 300 μm h$^{-1}$ was obtained using a low-pressure (<0.10 atm) HVPE system[13], but high-efficiency devices incorporating material grown at that rate were not demonstrated. Unlike in atmospheric-pressure HVPE, the general growth reaction of GaAs at low pressure involves uncracked $AsH_3$ instead of $As_2/As_4$ species because the number of collisions that $AsH_3$ molecules experience is relatively small at low pressure and a substantial fraction of $AsH_3$ is not cracked completely. This means the use of a low pressure HVPE system allows access to the hydride-enhanced mechanism[14] to augment the growth rate. In the hydride-enhanced growth mechanism, growth occurs mainly from uncracked $AsH_3$ with much faster kinetics[14] whereas growth from $As_2/As_4$ requires the reduction of AsGaCl surface complexes by $H_2$ to form GaAs and HCl. This rate-limiting-step features a high kinetic barrier of ~200 kJ/mol[13] leading to much lower growth rates, and is much more likely to occur at atmospheric pressure due to the short mean-free path that aids the conversion of $AsH_3$ into $As_x$ species. Fast growth at atmospheric pressure is potentially more useful, however, because high vacuum conditions impose stricter design requirements on reactor materials and physical shape and are typically more expensive to operate than atmospheric pressure systems.

Increased growth rates are not useful, however, if they lead to diminished device performance. Of particular importance is the ability of HVPE to generate extremely high growth rates for III–V materials without sacrificing material quality. While increased throughput is desirable for reasons related to cost-reduction, there are potential issues with increased growth rates, including reports of concurrently increasing EL2 defect densities[4,15]. Here, we show GaAs and GaInP growth rates in excess of 300 and 200 μm h$^{-1}$, respectively, using atmospheric-pressure D-HVPE. We grow and evaluate the performance of GaAs solar cells shown schematically in Fig. 1b, grown at these GaAs growth rates showing minimal degradation in open-circuit voltage ($V_{OC}$) compared with devices grown using lower rates, using $V_{OC}$ as a proxy for overall material quality. We examine the effects of high growth rate on point defect formation and find that these defect concentrations increase only slightly with growth rate. We discuss the impacts of high growth rate on the potential throughput of D-HVPE devices in future production. We note that the growth conditions for the GaInP layers in the solar cell structures were kept constant in order to relate changes in device performance to changes in the GaAs material quality as much as possible. Furthermore, these relatively low GaInP growth rates are necessary in the cell structures to provide sufficient time to accommodate the gas flow changes needed for the different device layers due to the constraint of only two reactor chambers[16], a constraint that will

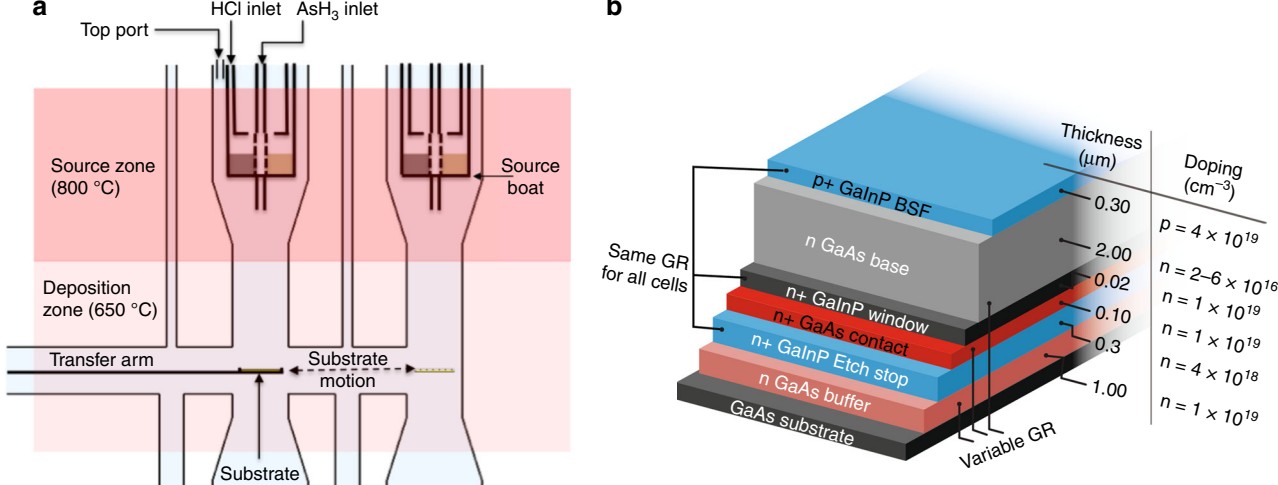

**Fig. 1** Schematics of D-HVPE reactor and GaAs solar cell. **a** Simplified schematic of the dual-chamber D-HVPE reactor used in this study. **b** Targeted device structure, grown in an inverted configuration, of the GaAs rear-heterojunction solar cells used in this study, identifying the layers in which the growth rate was varied

not be present in a future in-line reactor[17]. Thus, we did not incorporate the high growth rate GaInP into these devices in this work. However, we previously incorporated 54 µm h$^{-1}$ GaInP into two-junction GaInP/GaAs solar cells that showed $V_{OC}$ of 1.41 V despite an unpassivated structure[11] indicating a high quality GaInP cell can be grown at high growth rate. All solar cell growth rate discussions hereafter only concern the GaAs contact and base layers, as shown in Fig. 1b.

## Results

**Growth rate optimization**. For GaAs growth by D-HVPE, there are several factors that affect the growth rate in the mass-transport-limited HVPE growth parameter space. The first is the efficiency of the GaCl conversion reaction from HCl and Ga. This is governed in large part by the temperature in the source region, 800 °C in our case, but also by the residence time of the HCl in the Ga boat if the kinetics of the HCl to GaCl conversion reaction are not sufficiently fast. The HCl residence time in the Ga source boat is defined predominantly by the flow of H$_2$ carrier gas that pushes the HCl through the boat, $Q_{H_2}^{Ga}$. Figure 2a shows that reducing the H$_2$ flow rate through the Ga boat is a suitable method for dialing in specific growth rates over a wide range. The reduced carrier flow permits the generation of more GaCl for a given HCl flow rate and thus minimizes the amount of free HCl in the reactor. This free HCl would otherwise suppress the growth rate by driving the reverse of the GaAs growth reaction (i.e., etching of GaAs). In the case of the data shown in Fig. 2a for the

growth of GaAs, all other growth parameters, i.e., the substrate temperature, the HCl flow rate through the Ga boat, the AsH$_3$ partial pressure $\left(P_{AsH_3}\right)$, and the total system H$_2$ carrier flow $\left(Q_{H_2}^{Total}\right)$ of 9000 sccm, were held constant. Note that as $Q_{H_2}^{Ga}$ was reduced the H$_2$ was replaced in the top reactor port (see Fig. 1a) to maintain a constant total H$_2$ flow rate. Decreasing $Q_{H_2}^{Ga}$ from 2000 to 75 sccm increased the GaAs growth rate from about 50 to 100 µm h$^{-1}$. The increase in the growth rate is attributed to an increase in the conversion of Ga to GaCl as $Q_{H_2}^{Ga}$, and by extension, gas velocity, through the Ga boat is reduced.

The second factor affecting the GaAs growth rate is the mass transport of reactants to the growth surface. Figure 2b shows the effect of increasing the GaCl partial pressure by varying the HCl flow rate through the Ga boat ($P_{HCl(Ga)}$), using the $Q_{H_2}^{Ga}$ of 75 sccm condition discussed above. The red data in Fig. 2b, which use a H$_2$ carrier gas flow through the AsH$_3$ inlet (see Fig. 1a), $Q_{H_2}^{AsH_3}$ of 3125 sccm, show an initial increase in growth rate as the GaCl partial pressure increases from 0.8 to $1.7 \times 10^{-3}$ atm. However, for higher GaCl partial pressures the growth rate tends to stagnate and then slightly decrease. This may indicate that the growth rate is limited by a process whereby GaCl complexes compete with As species for adsorption sites at higher GaCl partial pressure[18]. The third factor affecting the GaAs growth rate is the delivery of uncracked AsH$_3$ to the growth surface. Delivering uncracked AsH$_3$, achieved by increasing the AsH$_3$ carrier flow rate, was

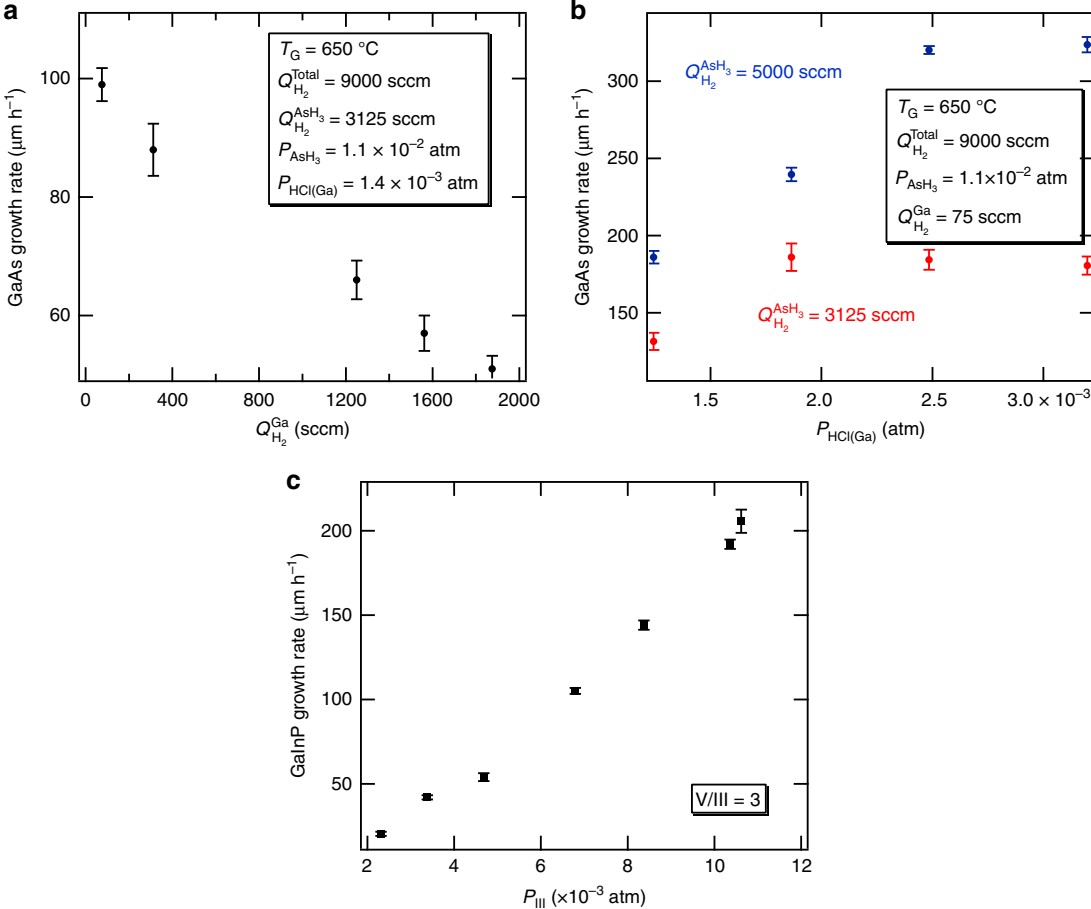

**Fig. 2** Effect of growth parameters on growth rate. GaAs growth rate as a function of H$_2$ carrier flow rate through the Ga boat (**a**) and partial pressure of HCl(Ga) sent to the boat (**b**). Other growth parameters indicated in each figure and the total reactor flows were held constant and (**c**) growth rate of GaInP as a function of group III partial pressure by D-HVPE. Error bars are standard deviations of growth rate measured from multiple positions on the samples

previously shown to increase GaAs growth rates[14]. Figure 2b shows that increasing $Q_{H_2}^{AsH_3}$ to 5000 sccm increases the growth rate for a given partial pressure of GaCl (compare red and blue data). Again, note that as $Q_{H_2}^{AsH_3}$ increased, $Q_{H_2}^{Total}$ was maintained at a constant level by removing an equal amount of $H_2$ flow from the top port. A linear increase in the growth rate is observed for a larger range of GaCl partial pressure even though a similar growth rate saturation seems to start at the highest $P_{HCl(Ga)}$. Combining these three effects, we achieved a maximum GaAs growth rate of higher than 320 $\mu m\,h^{-1}$ at a GaCl partial pressure of $2.4 \times 10^{-3}$ atm. We expect that continued increases in either the carrier gas flow rate or the $AsH_3$ partial pressure will further increase the growth rate, but these data represent the maximum flows possible through our current mass flow controllers.

We applied similar principles to grow GaInP with enhanced growth rates. Figure 2c plots the growth rate of D-HVPE-grown GaInP as a function of the sum of the GaCl and InCl partial pressures ($P_{III}$), with a $PH_3$ carrier flow rate, $Q_{H_2}^{PH_3}$ of 2500 sccm. Each epilayer was lattice-matched to the GaAs substrate. The V/III ratio was held constant in this set of experiments to obtain lattice-matching, because variations in this parameter can influence the relative incorporation efficiencies of Ga and In in the solid[19]. We measure GaInP growth rates exceeding 200 $\mu m\,h^{-1}$, significantly higher than other reports in the literature, with no signs of growth rate saturation in the range of partial pressures tested. Similar to the GaAs growths, we expect GaInP growth is enhanced by uncracked $PH_3$ reaching the surface at the employed $Q_{H_2}^{PH_3}$. The absence of saturation in growth rate is perhaps a result of the fact that the increasing $PH_3$ flow, increased with $P_{III}$ to maintain a constant V/III ratio, enhances the growth rate. Furthermore, it is likely that a greater proportion of the $PH_3$ reaches the surface without thermally cracking into $P_2/P_4$ species due to the fact that this molecule is more thermally stable than $AsH_3$[20].

**Solar cell performance as a function of growth rate**. We grew a series of GaAs solar cells using D-HVPE with widely varying GaAs growth rates (35–309 $\mu m\,h^{-1}$) to evaluate the GaAs material quality as a function of growth rate. Again, we note that the growth rates for GaInP in these devices were kept relatively low, as described above, in order to only investigate effects of high growth rate GaAs on device performance. Figure 3a–d displays the open-circuit voltage ($V_{OC}$), fill factor (FF), short-circuit current density ($J_{SC}$), and efficiency ($\eta$) extracted from light current density–voltage (J–V) measurements for the full series of solar cells as a function of growth rate, using rates from 35 to 309 $\mu m\,h^{-1}$ on (100) GaAs substrates offcut 4° toward (111) B substrates, and from 84 to 200 $\mu m\,h^{-1}$ on 6°B and 9°B miscut substrates. We note that there was no attempt to show only the best result at each growth rate, which leads to some scatter in the data, and that none of these devices had an antireflection coating (ARC) applied. Figure 3a shows that the $V_{OC}$s for all of these devices fall in the range from 1.04 to 1.07 V. $V_{OC}$ is an excellent indicator of the quality of the active layers in a solar cell because it directly reflects negative contributions from non-radiative recombination[21]. Our devices are only 50–80 mV lower than record GaAs devices[1], although there is a slight reduction in $V_{OC}$ at the highest growth rates. We expect application of an ARC to add 10–15 meV to the $V_{OC}$.

We note that devices were not optimized at each growth rate, i.e., the same growth recipe was used for each of the samples shown in Fig. 3 up to 200 $\mu m\,h^{-1}$. The one change made in devices grown at rates higher than 200 $\mu m\,h^{-1}$ was to increase

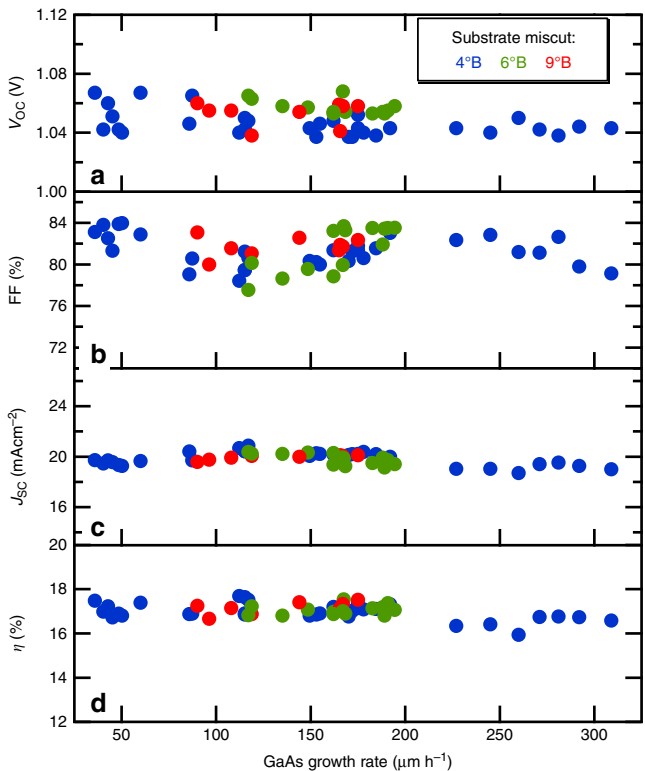

**Fig. 3** Performance of GaAs solar cells grown at growth rates from 35 to 309 $\mu m\,h^{-1}$. **a–d** Open-circuit voltage ($V_{OC}$), fill factor (FF), short-circuit current ($J_{SC}$) and efficiency ($\eta$), respectively, of solar cells grown at varying growth rates on different miscut (4°B, 6°B, and 9°B) substrates. All measurements were without antireflection coating and under the AM1.5 G illumination condition

the $H_2Se$ flow rate in the GaAs front contact layer in order to counteract reduced Se incorporation at these high growth rates (see Supplementary Fig. 1 and Supplementary Note 1). Efforts to optimize doping levels in other layers were not made. Although this approach led to very comparable results in terms $V_{OC}$, FF, $J_{SC}$, and $\eta$, we believe that continued optimization of the highest growth rate cells will lead to essentially identical performance for the entire range of growth rates. Figure 4 shows three NREL(National Renewable Energy Laboratory)-certified J–V measurements comparing GaAs solar cells grown with rates of 60, 195, and 292 $\mu m\,h^{-1}$ after application of ARCs. All devices achieved efficiencies of 24–25%. Figure 4d also shows the corresponding external quantum efficiency (EQE) spectra. The similar shapes of the EQE data confirm that the heterointerface quality is not degrading with growth rate. The change in the miscut angle only helped to enhance the dopant incorporation and improved the contact resistance (Supplementary Note 1) but does not otherwise affect device performance as can also be seen in Fig. 3. These results imply an insensitivity of the D-HVPE-grown material quality to growth rate over a large range. Nonetheless, it is possible that increases in point defects are affecting material quality at the highest growth rates. Recent high-growth-rate MOVPE studies reported sharp increases in deep-level defect concentrations, in particular the EL2 defect, with growth rate[4,15]. 'EL2' refers to a common GaAs mid-gap complex involving an As anti-site defect ($As_{Ga}$), which acts as a Shockley-Read-Hall lifetime limiting defect for minority carriers increasing recombination currents in solar cells and lowering device performance. We

**Fig. 4** Certified device performance of GaAs solar cells grown at high GaAs growth rates. *J-V* characteristics of 0.25 cm$^2$ single-junction GaAs cells with a MgF2/ZnS antireflection coating measured under the AM1.5G spectrum grown at growth rates of 60 μm h$^{-1}$ on 4°B miscut substrate (**a**), 195 μm h$^{-1}$ on 6°B miscut substrate (**b**), 292 μm h$^{-1}$ on a 6°B miscut substrate (**c**). **d** EQE spectra of the devices in (**a**) to (**c**)

performed deep-level transient spectroscopy (DLTS) measurements on GaAs devices grown by D-HVPE at rates from 60 to 309 μm h$^{-1}$ on 4°B miscut substrates and on devices grown at 180 μm h$^{-1}$ on 6°B and 9°B miscut substrates to understand the effect of growth rate on EL2 concentrations in HVPE. These devices have the same structure as the solar cells, with a n-GaAs/p$^{++}$GaInP heterojunction that was placed into reverse bias to measure trap densities. For the doping levels employed in this structure, the depletion region is almost entirely within the GaAs layer. We measured the doping density of each sample by the standard capacitance–voltage technique and used these values to compute a trap density[22]. In all D-HVPE-grown samples, only one trap was identified (see Supplementary Fig. 2 and Supplementary Note 2), with an activation energy of 0.82 eV, indicative of EL2[23,24].

Figure 5 shows the measured EL2 trap density for D-HVPE GaAs solar cells as a function of both growth rate and miscut angle. There is no clear trend with substrate miscut while

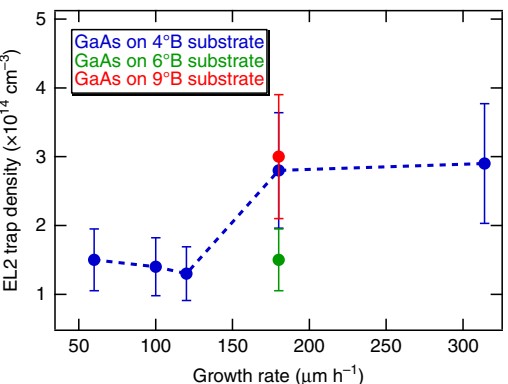

**Fig. 5** EL2 trap density as a function of growth rate for GaAs solar cells grown by D-HVPE. The error bars are standard deviations of trap density measured from the same sample multiple times

there appears to be a small increase in EL2 in the two highest growth rate samples. However, this factor of two increase is much smaller than results reported for MOVPE growth[15], where an order of magnitude increase in defect density was observed for GaAs grown at 640 °C and growth rates from 14 to 60 μm h$^{-1}$. We measured the magnitude of the capacitance transient as a function of the pulse width in the range 50–200 ns for the sample grown at 180 μm h$^{-1}$ on a 4°B substrate to determine the carrier capture rate, and calculated an electron capture cross-section of 3–4 × 10$^{-16}$ cm$^2$ and defect related lifetime of 220 ns which is longer than the calculated 125 ns radiative recombination lifetime of the n-GaAs base layer in our devices (Supplementary Fig. 3 and Supplementary Note 3). This electron capture cross-section is the same as that found by other groups in the literature for the EL2 defect in GaAs[24,25]. The observed trap levels are high enough that they may have a slight impact on $V_{OC}$[15], but it is clear that their effect is weak. The slightly lower $V_{OC}$ results shown in Fig. 3 for growth rates higher than 200 μm h$^{-1}$ might be a manifestation of higher point defect densities in these unoptimized structures.

## Discussion

Transferring these high growth rates to an industrial setting will have a beneficial impact on device growth times and overall throughput. In this section, we calculated the influence of growth rate on the total growth time of a single-junction GaAs solar cell, with the structure described in Fig. 1b, to provide insight into the benefit of high growth rates on overall device throughput. Here, we assume the modular in-line reactor described in ref. [6] in which each layer is deposited in a separate chamber as the wafer is shuttled through. Such a reactor would not have constraints that limit the growth rate of specific layers as in our dual-chamber reactor, and the in-line nature also minimizes the impact of heat-up and cool-down times. In an in-line system, each deposition zone would always be in steady state, and interfacial abruptness is controlled by the shape of the gas curtains[17]. This reactor does not yet exist, but the in-line concept is best able to make use of these high growth rates, and we wish to show growth times that may ultimately be achievable. We recognize that this design is perhaps aspirational, but similar in-line tools are in use for the deposition of CdTe solar cells. We calculated growth times using the range of growth rates demonstrated in this work, i.e., up to 320 μm h$^{-1}$ for GaAs and 206 μm h$^{-1}$ for GaInP. We also extended this analysis to the dual-junction GaInP/GaAs tandem structures demonstrated by D-HVPE in ref. [11] in order to show the effect of these growth rates on a device that contains a relatively thick GaInP layer. Figure 6a shows a contour plot of the total growth time for the single-junction GaAs solar cell, as a function of the GaAs and GaInP growth rates, while Fig. 6b shows the same plot for the GaInP/GaAs tandem. Here, we have neglected the transfer time between successive growth chambers. Transfer takes ~2 s in our current system, indicating that an additional 10–20 s of total process time may be required on top of the times listed in Fig. 6.

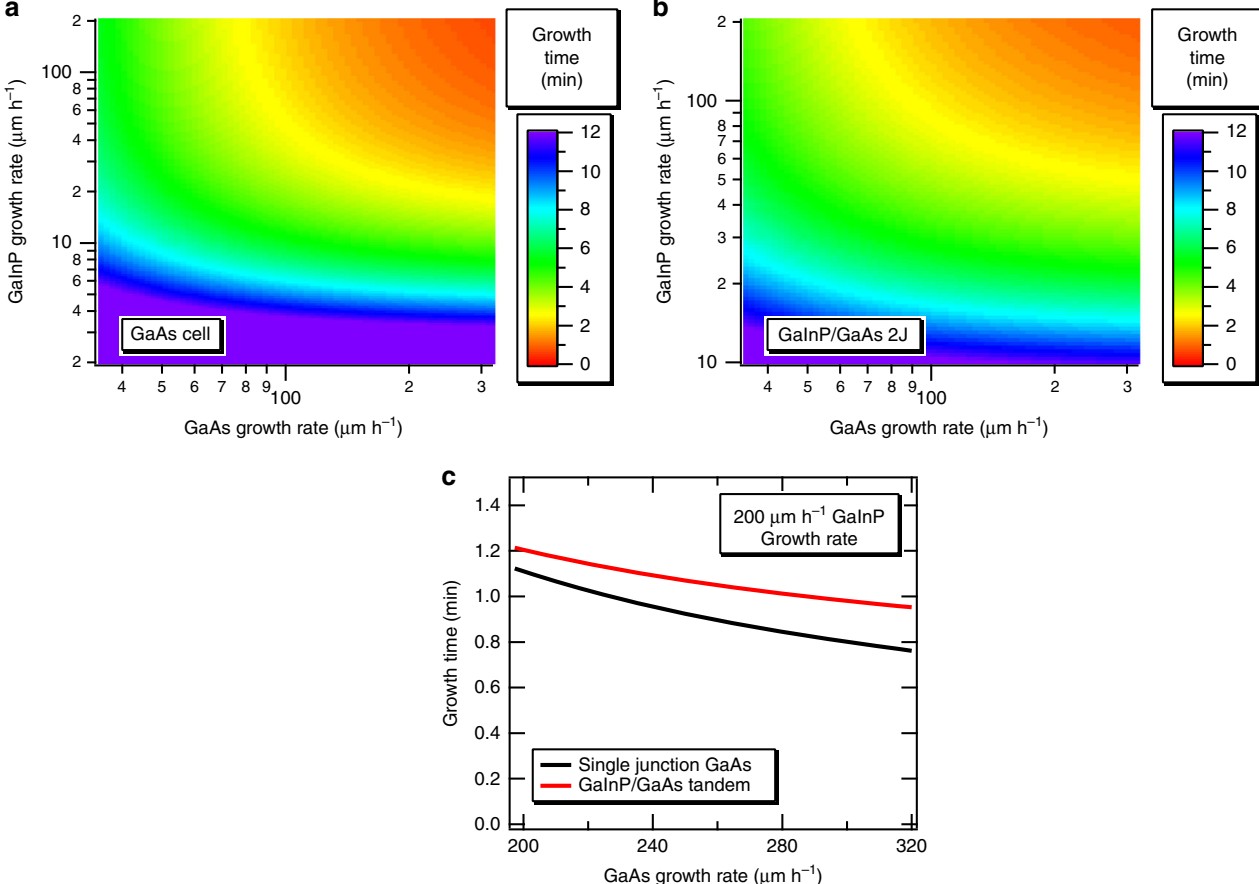

**Fig. 6** Impact of high growth rate on device throughput by a future optimized in-line HVPE reactor. **a** Total growth time for the single-junction GaAs solar cell structure shown in Fig. 1b as a function of the GaAs and GaInP growth rates, total growth time for a GaInP/GaAs two-junction solar cell structure (**b**), and slices through the data of Fig. 6b, c showing continued and significant growth time decreases at very high rates (**c**)

The main difference between the two structures is the inclusion of a GaInP base layer that is ~1-µm thick in the GaInP/GaAs tandem. The importance of high GaInP growth rates is thus heightened for this structure, making GaInP growth rates near those for GaAs necessary to maintain high overall throughput. Clearly there are dramatic reductions in growth time for the range of rates experimentally demonstrated in the current work with solar cell growth times plummeting to about 1 min for the highest rates. However, the device structure shown in Fig. 1b is also unoptimized, and we expect that several layers, including the buffer, etch stop, and base, can decrease in thickness without adversely affecting device performance. This would lead to growth times for single-junction GaAs solar cells that are well under the 1 min mark. Similar results are obtained for the GaInP/GaAs tandem, if GaInP growth rates near $200 \, \mu m \, h^{-1}$ are utilized. We have not explicitly confirmed the GaInP material quality at these very high rates, but we note that our previous GaInP/GaAs two-junction solar cells successfully used GaInP growth rates of $54 \, \mu m \, h^{-1}$ [11], and that significant degradation in GaInP quality would be in stark contrast to the GaAs solar cell data discussed above. These calculations show that a two-junction structure capable of achieving higher than 30% efficiency[11] could take about a minute to produce.

Increasing the growth rate continues to pay significant dividends on throughput, even at these very high rates. Figure 6c shows a slice through the data in the contour plots displayed in Fig. 6a, b, using our highest demonstrated GaInP growth rate of $200 \, \mu m \, h^{-1}$. Increasing the GaAs growth rate from 200 to $300 \, \mu m \, h^{-1}$ reduces the solar cell growth time by ~30% for both the single- and dual-junctions, representing a significant increase in throughput. The point of diminishing returns for high growth rates will certainly depend on the specific device structure, but there is clearly a benefit to high growth rates for solar cell devices. Other devices that require thick layers could benefit as well, including buried heterostructure quantum cascade lasers[26] and optical frequency conversion devices[27].

In summary, we report GaAs growth rates up to $320 \, \mu m \, h^{-1}$, and GaInP growth rates up to $206 \, \mu m \, h^{-1}$, by atmospheric-pressure D-HVPE at 650 °C. These rates are higher than previously achieved by low pressure, traditional HVPE. We obtained these enhanced growth rates by controlling the flow of hydrogen carrier gas in our system. The $V_{OC}$ of single-junction GaAs solar cells grown using GaAs growth rates from 35 to $309 \, \mu m \, h^{-1}$ was in the range of 1.04–1.07 V indicating low levels of non-radiative recombination regardless of growth rate. DLTS measurements identified only EL2 traps with concentration $< 3 \times 10^{14} \, cm^{-3}$ in this growth rate range, further corroborating the quality of D-HVPE devices grown at high rates. We showed that single-junction GaAs and dual-junction GaInP/GaAs solar cells can potentially be grown in <1 min when using the highest achieved growth rates of $320 \, \mu m \, h^{-1}$ for GaAs and $206 \, \mu m \, h^{-1}$ for GaInP. The growth of high quality III–V materials at these extreme growth rates can potentially lower deposition costs of high-efficiency optoelectronic devices that require thick layers.

## Methods

**D-HVPE III–V device growth**. All growth experiments were performed in atmospheric-pressure dual-chamber D-HVPE system[16] schematically shown in Fig. 1a. The sources used in our D-HVPE system are AsH$_3$ and PH$_3$ for the group V sources, and GaCl and InCl, which are formed in situ by flowing anhydrous HCl over Ga and In metal, as the group III sources. The dopants are Zn (p-type) and Se (n-type) supplied as diethylzinc and H$_2$Se, respectively. The source zone where the metal chlorides are formed is held at 800 °C, while the deposition zone (growth temperature, $T_G$) is held at 650 °C for all materials grown in this work. Growth rate studies were conducted by growing lattice-matched GaAs/GaInP/GaAs structures on (100) GaAs substrates miscut 4° towards (111)B. Epilayer thickness was

determined by selectively etching a portion of the GaAs or GaInP layer to a GaInP or GaAs etch stop, respectively, and measuring the step height with stylus profilometry. Growth rates were calculated from these thicknesses and the known growth time. Every individual solar cell growth rate reported below was measured in this fashion. Changes in growth rate were studied as functions of gas flow rates in the reactor, including HCl, AsH$_3$, and H$_2$ carrier gas injected into different parts of the system. In addition, single-junction GaAs solar cells were grown in an inverted configuration with lattice-matched GaInP window and back surface field (BSF) layers in a rear-heterojunction design[28]. The structure of the solar cell used in this work is shown in Fig. 1b along with the targeted thickness and doping of each layer. Solar cells were grown on (100) GaAs substrates offcut either 4°, 6°, or 9° toward (111)B at growth rates from 35 to $309 \, \mu m \, h^{-1}$. V/III ratios for the GaAs layers ranged from 4 to 7. GaInP layers in the solar cell structures were grown at $2.3 \, \mu m \, h^{-1}$ for the etch stop and window and $6 \, \mu m \, h^{-1}$ for the BSF layers and were the same for all solar cells in this study.

**Solar cell processing**. Device processing proceeded as previously described[9]. Au was electroplated on the BSF, serving as a back contact as well as a back reflector to increase the optical path length of the device. This Au back contact was bonded to a Si handle using epoxy and the GaAs substrate and GaAs:Se buffer layer were removed by a selective chemical etch that stops at the GaInP etch stop layer. The front Au grid formation and $0.25 \, cm^2$ square mesa isolation for solar cells and $0.7 \, mm^2$ rectangular arrays for transmission line measurements were completed using standard photolithography techniques. On select samples, a MgF$_2$ (100 nm)/ZnS (52 nm) antireflection coating (ARC) was deposited in a thermal evaporator.

**Solar cell characterization**. We measured solar cell external quantum efficiency (EQE) on a custom instrument in which chopped, monochromatic light was split and then sent to the device and a calibrated, broadband reference diode. We measured the output current of the device and reference on a lock-in amplifier, and used it to calculate the EQE, which is the ratio of electron current out to incident photons absorbed. Specular reflectance from the device surface was measured with a separate, calibrated reference diode.

We compared the measured EQEs with GaAs reference cell to calculate spectral correction factors for the AM1.5G spectrum. We set the height of a Xe-lamp solar simulator to obtain an illumination of $1000 \, W/cm^2$, determined by measuring the current from the reference GaAs held under the lamp and adjusting by the spectral correction factor. The cell one-sun $J$–$V$ curves were measured under the adjusted spectrum.

**Deep-level transient spectroscopy studies**. Deep-level transient spectroscopy (DLTS) measurements were performed on select samples to study the effect of growth rate on the trap type and density. Our DLTS system uses Fourier transforms to characterize full capacitance transients, and we used a reverse bias voltage of $−5.0$ V, a trap filling pulse of 0.70 V and a pulse width of 1.0 ms for these measurements[29]. The DLTS measurements were performed on $0.7 \, mm^2$ inverted devices after substrate removal with the same structure as the solar cells.

**Reporting summary**. Further information on research design is available in the Nature Research Reporting Summary linked to this article.

## Data availability

The data that support the findings of this study are available from the corresponding author upon reasonable request.

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

## Acknowledgements

The authors would like to thank David Guiling for HVPE materials growth, and Michelle Young and Daniel Lu for device processing. This work was authored by Alliance for Sustainable Energy, LLC, the manager and operator of the National Renewable Energy Laboratory for the U.S. Department of Energy (DOE) under Contract No. DE-AC36-08GO28308. This material is based upon work supported by the U.S. Department of Energy's Office of Energy Efficiency and Renewable Energy (EERE) under Solar Energy Technologies Office (SETO) Agreement Number 30290. The views expressed in the article do not necessarily represent the views of the DOE or the U.S. Government. The U.S. Government retains and the publisher, by accepting the article for publication, acknowledges that the U.S. Government retains a nonexclusive, paid-up, irrevocable, worldwide license to publish or reproduce the published form of this work, or allow others to do so, for U.S. Government purposes.

## Author contributions

W.M. designed the experiments, characterized the devices, analyzed the data, and wrote the paper. K.L.S. helped in the design of growth experiments and analyzing the data. J.S. helped in the design of the experiments and analyzing the data. S.J. performed deep-level transient spectroscopy measurements on the samples. A.J.P. helped in the design of the experiments and data analysis. All authors contributed to the writing in this paper.

## Additional information

**Competing interests:** The authors declare no competing interests.

