## [Peer Review File · Nature Communications]

Reviewers' comments:

Reviewer #1 (Remarks to the Author):

This study presents recent developments in D-HVPE growth of GaAs based rear-heterojunction solar cells, which are certainly of interest for the PV community, and beyond. The focus is on growth parameters influence for growth rate and material quality. The authors demonstrate that low EI2 trap density can be maintain at low level for growth rate up to $\sim 300 \mu\text{m/h}$, for GaAs layers. Excellent cells results are shown with 25% Eff for $\sim 200 \mu\text{m/h}$ growth rate. This manuscript can be improved by addressing the different points listed below. Overall it would be valuable to discuss more about the global perspectives such as large wafer/cell size, homogeneity, growth rate of indium containing layers, interface abruptness, TJ, 2/3J cells, etc.

- Concerning the cost discussion in the introduction, It would be usefull to remind which fraction of the cell cost epitaxy does represent, as well as the fraction of the cell cost compared to system level cost. This would help the ready to quantify this cost reduction potential
 - Mention the reactor pressure in the experimental details (before the LPCVD GR comparison)
 - Choice of determination method for growth rate: can you comment why profilometry was chosen ? Did you correlate with optical or XRD methods ? Is it possible to follow the real time growth rate in-situ with this type of D-HVPE reactor ?
 - Do you perform statistics at different spots for layer thickness measurement ?
 - It would be interesting to show total cell growth time as a function of GaAs epitaxial growth rate, to better understand the influence of this parameter given that fixed growth (& low) rate is used for GaInP layers.
 - Which growth rate upper limit of Indium or Gallium containing layers can you expect?
 - Fig.2: error bars would be great. The 3125 sccm curve is red (not green as described in the text)
 - There is no conclusion for the discussion on contact layer doping ajustement, what seems to be the best approach and why ?
- To illustrate defect concentration increase with growth rate, the reference to schmieder et al. can also be completed with a reference to the study on diffusion length performed by Lang et al. cited in the article beginning (Ref 4).
- It would be great to have quantitative meas. and/or literature comparison of EI2 trap density impact on lifetime/diffusion length or cells parameters.
 - When comparing growth rate, more details are needed around this x20 factor: total cell growth time comparison in both scenarios, as well as comments on why GaInP GR are not pushed further yet.
 - Fig. 5: do you also have statistical date on the same wafer ? how about spatial uniformity of GR, defects and performances ?
 - I would moderate a bit the last sentence to say that GR has very little influence on cell performance.
 - It would be interesting to have quantitative meas. and/or calculations of precursor utilization together with MOCVD comparison.
 - It would also be usefull to use the Woc values in this paper

Reviewer #2 (Remarks to the Author):

This paper represents characterization of the effect of growth rate on GaAs solar cells grown by HVPE. The paper achieved an extremely high growth rate of $\sim 300 \mu\text{m/h}$ and a first-ever demonstration of GaAs solar cells grown at such a high growth rate with maintaining superior conversion efficiencies by atmospheric pressure HVPE. I think the paper may impact on both the condensed matter science and

industrial fields. The paper is well-organized and clear. The paper deserves to be published in the Nature communications, though there are several areas that I think it should be addressed before possible publication as listed below.

- 1) In line 123 of page 6, the "green data" may be "red data". Please recheck it.
- 2) In line 137 of page 7, did you evaluate the actual values of source utilization efficiency?
- 3) In line 141 of page 7 and fig. 2(b), the saturated growth rate of ~ 320 $\mu\text{m}/\text{h}$ in your results was fortuitously similar with ref. 7. Was ~ 300 $\mu\text{m}/\text{h}$ a theoretical limitation in HVPE growth? Or is there any possibility to surpass it by further modifying the growth parameter?
- 4) I think it becomes easy for us to follow your growth parameter (each carrier flow) if you show the schematic structure of the D-HVPE reactor.
- 5) In line 149 of page 7 and Fig. 3(a), please give us the information about the growth rate for the n-GaAs base layer.
- 6) In fig.3 (b), I would like to know how you maintained the doping density of the n-GaAs base layer to $4 \times 10^{16} \text{ cm}^{-3}$. Did you control the H_2Se flow for each cell?
- 7) In line 176 of page 9, I agree with your discussion that higher miscut angle enhances the doping density. On the other hand, was there any difference on the surface roughness or morphology between the low and very high growth rates? I think the surface roughness may affect the incorporation efficiency of the dopants. In addition, did the miscut affect the material quality of InGaP layers?
- 8) In fig. 5, the J_{sc} was successfully maintained over the wide range of the growth rates. Was there any difference in EQE spectra between a low and a high growth rate? I think the EQE spectra is one of the metrics to compare the diffusion length of carriers in the base layer and these may strengthen the conclusions.

Reviewer #3 (Remarks to the Author):

The manuscript of Metaferia et al. describes experiments performed towards high growth rate HVPE of GaAs for solar cell applications. One novelty which is claimed is the growth of GaAs at up to $300 \mu\text{m}/\text{h}$ using an atmospheric pressure reactor rather than a low pressure reactor. The specific influence of pressure is not discussed and high growth rate HVPE of GaAs is not really new. Also the authors have previously reported GaAs solar cells grown by HVPE at $60 \mu\text{m}/\text{h}$ with efficiencies of 20.6%. Now they have increased the growth rate to $195 \mu\text{m}/\text{h}$ for a cell with 25% conversion efficiency which is a great result and I want to congratulate the authors to this achievement and the good work they have published in the field of HVPE growth in recent years! Still the new reported efficiency value is significantly below the world record GaAs solar cell with 29.1% efficiency under the same AM1.5g conditions. Certainly the new result of Metaferia et al is the best GaAs solar cell grown by HVPE at such a high growth rate. These results deserve to be published and are of interest to the III-V solar cell community. But beyond the III-V solar cell community there are little applications which benefit from such high GaAs growth rates. I am therefore not sure if this work is suitable for Nature Communications or should rather be published in a solar cell journal like IEEE JPV or Progress in Photovoltaics.

The manuscript is well organized, the length is appropriate to the content and the results well described. Still I believe that some improvements should be done:

- discussion of Fig.2 refers to green data but the data shown is red.
- I do not understand how a higher H₂ carrier gas flow which is injected together with the same amount of AsH₃ changes the amount of H radicals in the reactor. There is not more AsH₃ and why should this influence the hydrogen radical concentration in the reactor? I would recommend further experiments to clarify this influence of carrier gas flow on the growth rate of GaAs. Was the total H₂ flow through the HVPE machine kept constant in these experiments? What happens if the H₂ carrier gas is increased even further? Is there a jetting inside the reactor? Is it possible that the AsH₃ distribution in the gas phase is changing? Maybe perform some modelling of the gas inside the reactor as a function of the carrier gas flow rates?
- Another important finding of the paper in Figure 4 is that the GaAs EL2 trap density is more or less constant with growth rate and that this result differs from MOVPE growth reports where EL2 density increased significantly with growth rate. I believe that this result requires more discussion and maybe also additional experiments. It is unclear to the reviewer why there should be a difference between HVPE and MOVPE for the same V/III ratios? Both processes are gas phase processes and the EL2 should be a result of excess As on the surface which incorporates with higher probability as antisite if the growth rate is increased. I do not understand why HVPE would not show this effect. Therefore, the authors should discuss V/III ratio variation, influence of reactor pressure (?) and other possible influences on the observed EL2 density. V/III ratios should be given for the MOVPE and HVPE experiments. Also Figure 4 shows some increase in EL2 density from 100 to 200 $\mu\text{m}/\text{h}$ and all HVPE results are above the low growth rate MOVPE material. I believe that this requires more discussion.
- some references did not display correctly on my computer and there is a Mm/h in Ref. (4) which should be changed to $\mu\text{m}/\text{h}$

Reviewer #1

This study presents recent developments in D-HVPE growth of GaAs based rear-heterojunction solar cells, which are certainly of interest for the PV community, and beyond. The focus is on growth parameters influence for growth rate and material quality. The authors demonstrate that low EL2 trap density can be maintain at low level for growth rate up to $\sim 300 \mu\text{m/h}$, for GaAs layers. Excellent cells results are shown with 25% Eff for $\sim 200 \mu\text{m/h}$ growth rate. This manuscript can be improved by addressing the different points listed below.

1. *Overall it would be valuable to discuss more about the global perspectives such as large wafer/cell size, homogeneity, growth rate of indium containing layers, interface abruptness, TJ, 2/3J cells, etc.*

We have expanded our discussion in the manuscript and included more references on the perspectives of our dual chamber dynamic HVPE to grow complex structures such as tunnel junctions (A.J. Ptak *et al.*, in IEEE Journal of Photovoltaics, 8(1), pp. 322-326.), and GaInP/GaAs tandem solar cells (K.L. Schulte, *et al.*, Prog Photovolt Res Appl 2018;26:887–893). Both kinds of devices require an accurate control of both material and doping concentrations as well as interface abruptness in terms of both doping and materials. These devices, difficult to grow by traditional HVPE, are possible because of the rapid back and forth movement of the substrate for the growth in the two chambers. Each chamber of the dual chamber dynamic HVPE is capable of growing binary, ternary and quaternary compositions in the entire GaInAsP family of materials, which provides flexibility in the realization of many III-V devices including multi-junction solar cells. We have not shown large area growth results in this manuscript, however literature studies showed the feasibility of HVPE to grow large area uniform high quality films, for example, N. Liu, *et al.* in Journal of Crystal Growth 388 (2014) 132–136 showed the growth of highly uniform GaN layer on a 2-inch wafer with thickness variations of $< 4\%$ and Kyma Technologies, Inc. has demonstrated the growth of GaN thin films and free standing substrates up to 8 inch diameter (Leach *et al.*, in proceeding of Compound Semiconductor Manufacturing Technology (CS MANTECH), 2018)). We have added these references and related discussion to the manuscript.

Although we didn't originally show our growth rate studies for In containing layers in our manuscript, we agree with the reviewer that it is important to understand how the growth rate of In-containing layers affects overall device throughput. In the manuscript, we now show that we have grown GaInP films at growth rates up to $\sim 206 \mu\text{m/h}$, without obvious saturation of the rate using the growth parameters possible with our existing equipment. We also point out that the GaInP in our published GaInP/GaAs tandem cells were also grown at $54 \mu\text{m/h}$ (K.L. Schulte, *et al.*, Prog Photovolt Res Appl 2018;26:887–893). We believe the growth rate of

GaInP can be pushed further by continued optimization of our growth conditions and have not yet encountered any physical limitation in achieving high growth rates, similar to our GaAs films. We added a new section discussing this in the manuscript.

2. Concerning the cost discussion in the introduction, It would be use full to remind which fraction of the cell cost epitaxy does represent, as well as the fraction of the cell cost compared to system level cost. This would help the ready to quantify this cost reduction potential

A detailed discussion of the cost of HVPE-grown III-V photovoltaic devices is presented by Simon et al., cited as Ref. 6 in this manuscript. We also included a reference on techno-economic analysis and cost reduction roadmap for III-V solar cells by Horowitz et al. However, it is difficult to quantify the true cost reduction potential because cell cost depends on numerous other factors besides epi growth rate, many of which are the subject of intense research. The marginal benefit of increased growth rate will asymptote at a specific value when these other costs are fixed, but if these can come down the impact of growth rate will be elevated again. Our main focus in this work is to understand the effects of extremely high growth rates on device efficiency and to push the limits of this technology, so as to provide information readers can use when doing cost modeling for their own specific application.

3. Mention the reactor pressure in the experimental details (before the LPCVD GR comparison)

Our reactor operates at atmospheric pressure. We added this information in the experimental section of the manuscript as suggested by the reviewer.

4. Choice of determination method for growth rate: can you comment why profilometry was chosen? Did you correlate with optical or XRD methods? Is it possible to follow the real time growth rate in-situ with this type of D-HVPE reactor?

Given that ultra-selective etchants exist for III-As against III-P materials, we find stylus profilometry of etched steps to be significantly more accurate than x-ray diffraction or post-growth optical measurements. The technique is fast and reliable. Currently, we do not have a real-time, *in-situ*, growth rate monitoring system in our HVPE reactor, but there is nothing physically preventing the use of *in-situ* reflectance (and similar) methods that are often used in MOVPE systems.

5. Do you perform statistics at different spots for layer thickness measurement

Yes, we have measured at multiple spots and taken the average thickness, and we now include error bars (standard deviation) in Fig. 2.

6. It would be interesting to show total cell growth time as a function of GaAs epitaxial growth rate, to better understand the influence of this parameter given that fixed growth (& low) rate is used for GaInP layers.

We thank the reviewer for this suggestion, and agree that this is a very useful addition, especially given the new data that we are including on GaInP growth rates. We added a new section discussing the influence of GaAs growth rate on the total growth time of single- and dual-junction GaAs solar cells. We also discussed the influence of GaInP growth rate on the total cell growth time.

7. Which growth rate upper limit of Indium or Gallium containing layers can you expect?

Please see our reply to points 1 and 6 above. As shown in the manuscript, we do not currently see any physical limit to growth rate for either GaAs or GaInP – crucially, while maintaining excellent material quality – up to the limit of our reactor's current equipment capabilities.

8. Fig.2: error bars would be great. The 3125 sccm curve is red (not green as described in the text)

We added error bars to Fig. 2 (please also see our response to point 5 above). We thank the reviewer for catching our mistake re: the color of the curve, and we have corrected this in the text.

9. *There is no conclusion for the discussion on contact layer doping adjustment, what seems to be the best approach and why?*

We hesitate to draw a strong conclusion given that all three approaches were successful at improving the contact resistance. Because all three approaches work equally well, the “best” solution may be application specific. However, to maintain maximum throughput, in keeping with the spirit of this manuscript, increasing the H₂Se flow appears to be the simplest solution. We have added a sentence in the manuscript to highlight this point.

10. *To illustrate defect concentration, increase with growth rate, the reference to schmieder et al. can also be completed with a reference to the study on diffusion length performed by Lang et al. cited in the article beginning (Ref 4).*

We thank the reviewer for pointing out this, we now added this reference in the manuscript accordingly.

11. *It would be great to have quantitative meas. and/or literature comparison of EL2 trap density impact on lifetime/diffusion length or cells parameters.*

We agree with the reviewer that it is very important to understand the quantitative effect of EL2 trap density on solar cell performance. The difficulty is in choosing the proper capture cross-section to use in the calculations. It is not clear to us that the infinite-temperature capture cross-section derived from the intercept of an Arrhenius plot is appropriate for the calculation of lifetime at room temperature. However, in order to provide a general idea of whether or not EL2 is limiting our lifetime, we used electron and hole capture cross-sections for MOVPE grown GaAs from the literature (M. A. Zaidi et al., in Appl. Phys. Lett. 61, 2452 (1992)) and calculated the Shockley-Read-Hall (SRH) lifetime of 1215 ns and 609 ns for our devices grown at 60 and 309 $\mu\text{m/h}$, respectively. The corresponding EL2 trap densities from these devices are $1.5 \times 10^{14} \text{ cm}^{-3}$ and $3 \times 10^{14} \text{ cm}^{-3}$. These values of SRH lifetime are much longer than the calculated radiative lifetime of 125 ns, indicating our devices are likely not limited by EL2 traps. However, we don't wish to include these data in the manuscript because they are based on the assumption that the capture cross-sections from MOVPE- and HVPE-grown GaAs are the same, and that this cross-section is the proper value to use in these calculations.

12. *When comparing growth rate, more details are needed around this x20 factor: total cell growth time comparison in both scenarios, as well as comments on why GaInP GR are not pushed further yet.*

The 20x factor was estimated based on the demonstrated 309 $\mu\text{m/h}$ HVPE growth rate and a “typical” MOVPE growth rate. We understand that recent efforts by Lang, *et al.*, Schmieder, *et al.*, and Sodabanlu, *et al.* are making strides toward producing high-efficiency devices at higher GaAs growth rates using MOVPE, but we do not believe that these conditions are currently standard practice. Please see our reply to points 1, 6 and 7 for discussion of the total cell growth time comparison at different growth rates.

13. *Fig. 5: do you also have statistical data on the same wafer? how about spatial uniformity of GR, defects and performances?*

The focus of this manuscript is to investigate the growth rate ceiling of D-HVPE and assess the material quality at high growth rates. To efficiently maximize our resources, we studied relatively small devices (0.25 cm^2) that we believe are quite uniform over this area. We agree with the reviewer that uniformity studies are essential and plan to investigate this factor in subsequent work.

14. *I would moderate a bit the last sentence to say that GR has very little influence on cell performance.*

We agree with the reviewer that the last sentence should be moderated as suggested and we changed the last sentence in the manuscript accordingly.

15. *It would be interesting to have quantitative meas. and/or calculations of precursor utilization together with MOCVD comparison.*

We agree with the reviewer on this point, but feel it is not possible to compare results obtained in a commercially available MOVPE reactor with results from our research reactor. We have estimated up to 70% Ga utilization in our research reactor over small areas, but without a more rigorous study of utilization over larger areas in a commercially optimized reactor, we do not feel comfortable speculating about utilization.

16. *It would also be useful to use the W_{OC} values in this paper*

Although many reports in the literature use W_{OC} as a metric for general material quality assessment in heterostructure devices, it is our opinion that W_{OC} should only be used for homojunction devices for which the band gap energy of the device is clearly defined. Our GaAs cells are heterostructure devices, and the definition of band gap for these structures is unclear. Therefore, we haven't included W_{OC} and instead use V_{OC} as a proxy for the material quality of our devices. For reference, however, the extracted W_{OC} values for our devices range from 0.34 – 0.37 V, if we assume that the relevant band gap in these devices is that of GaAs, and there is no adjustment necessary from the GaInP side of the junction.

Reviewer #2 (Remarks to the Author):

This paper represents characterization of the effect of growth rate on GaAs solar cells grown by HVPE. The paper achieved an extremely high growth rate of ~300 $\mu\text{m}/\text{h}$ and a first-ever demonstration of GaAs solar cells grown at such a high growth rate with maintaining superior conversion efficiencies by atmospheric pressure HVPE. I think the paper may impact on both the condensed matter science and industrial fields. The paper is well-organized and clear. The paper deserves to be published in the Nature communications, though there are several areas that I think it should be addressed before possible publication as listed below.

1) *In line 123 of page 6, the “green data” may be “red data”. Please recheck it.*

We thank the reviewer for also noticing our mistake, which we have corrected.

2) *In line 137 of page 7, did you evaluate the actual values of source utilization efficiency?*

Please see our response to Reviewer #1, point 15 above. We haven't evaluated the actual values of source utilization efficiency. However, we note that our growth rate as a function of carrier gas flow shows that our growth rate increases without a change in the input partial pressure of the sources, indicating increased material utilization as growth rate increases.

3) *In line 141 of page 7 and fig. 2(b), the saturated growth rate of ~320 $\mu\text{m}/\text{h}$ in your results was fortuitously similar with ref. 7. Was ~300 $\mu\text{m}/\text{h}$ a theoretical limitation in HVPE growth? Or is there any possibility to surpass it by further modifying the growth parameter?*

To our knowledge, ~ 300 $\mu\text{m}/\text{h}$ is not a limit for the HVPE growth rate, merely a coincidence between the two studies. In fact, our results in Fig. 2(b) indicate that it may be possible to further increase the growth rate by increasing either the AsH_3 flow or the H_2 carrier gas flow rates. Unfortunately, we maxed out the flows for each parameter in our reactor, so we could not push the growth rate further. We have noted this fact in the manuscript.

4) *I think it becomes easy for us to follow your growth parameter (each carrier flow) if you show the schematic structure of the D-HVPE reactor.*

We agree with the reviewer that adding reactor schematic could make our explanation clearer and we added a simplified schematic of our dual chamber dynamic HVPE reactor as Figure 1 (a).

5) *In line 149 of page 7 and Fig. 3(a), please give us the information about the growth rate for the n-GaAs base layer.*

We thank the reviewer for pointing out this oversight. We added the growth rate of the base layer in the caption of Fig. 3(a).

6) *In fig.3 (b), I would like to know how you maintained the doping density of the n-GaAs base layer to $4 \times 10^{16} \text{ cm}^{-3}$. Did you control the H_2Se flow for each cell?*

As discussed in the manuscript, our devices grown at growth rates $> 200 \text{ } \mu\text{m/h}$ were grown with higher H_2Se flow in order to counteract the decrease in Se incorporation at higher growth rates. Even though our targeted doping concentration in the base is $4 \times 10^{16} \text{ cm}^{-3}$, there is some variation in the GaAs doping density from sample to sample due to the changing growth rates. The doping concentration from capacitance voltage measurements of our cells ranged from $2 - 6 \times 10^{16} \text{ cm}^{-3}$, close to the target of $4 \times 10^{16} \text{ cm}^{-3}$. We do not believe this variation significantly affects the results, but we now note in the text that we did not specifically control the doping density from sample to sample.

7) *In line 176 of page 9, I agree with your discussion that higher miscut angle enhances the doping density. On the other hand, was there any difference on the surface roughness or morphology between the low and very high growth rates? I think the surface roughness may affect the incorporation efficiency of the dopants. In addition, did the miscut affect the material quality of InGaP layers?*

We have no evidence that the miscut angle affected the surface roughness or quality of the GaInP in a substantial way that would impact solar cell performance.

8) *In fig. 5, the J_{sc} was successfully maintained over the wide range of the growth rates. Was there any difference in EQE spectra between a low and a high growth rate? I think the EQE spectra is one of the metrics to compare the diffusion length of carriers in the base layer and these may strengthen the conclusions.*

There is no significant difference in the EQE spectra between cells grown at high and low growth rates. We added a figure showing EQE spectra from three cells grown at rates of 60, 195, and 292 $\mu\text{m/h}$ in the manuscript as Fig. 6(d) to confirm that our interfaces are not degrading with growth rate. The EQE data are very similar, so we cannot say unambiguously that the diffusion length is not changing with growth rate. We can only say that changes in diffusion length, if there are any, are not large enough to limit device performance.

Reviewer #3 (Remarks to the Author):

The manuscript of Metaferia et al. describes experiments performed towards high growth rate HVPE of GaAs for solar cell applications. One novelty which is claimed is the growth of GaAs at up to 300 $\mu\text{m/h}$ using an atmospheric pressure reactor rather than a low-pressure reactor. The specific influence of pressure is not discussed and high growth rate HVPE of GaAs is not really new.

We thank the reviewer for pointing this out. We added the following text in the manuscript to clarify the effect of pressure on the growth mechanism and show that the hydride enhanced mechanism we explained in the manuscript enables us to achieve in an atmospheric pressure system what can previously only be achieved in a low-pressure system.

“... Unlike in atmospheric-pressure HVPE, the general growth reaction of GaAs at low pressure involves uncracked AsH_3 instead of As_2/As_4 species because the number of collisions that AsH_3 molecules experience

is relatively small at low pressure and a substantial fraction of AsH₃ is not cracked completely. This means the use of a low pressure HVPE system allows access to the hydride enhanced mechanism to enhance the growth rate. Fast growth at atmospheric pressure is potentially more useful, however, because high vacuum conditions impose stricter design requirements on reactor materials and physical shape, and are typically more expensive to operate than atmospheric pressure systems”

Also the authors have previously reported GaAs solar cells grown by HVPE at 60 μm/h with efficiencies of 20.6%. Now they have increased the growth rate to 195 μm/h for a cell with 25% conversion efficiency which is a great result and I want to congratulate the authors to this achievement and the good work they have published in the field of HVPE growth in recent years! Still the new reported efficiency value is significantly below the world record GaAs solar cell with 29.1% efficiency under the same AM1.5g conditions. Certainly, the new result of Metaferia et al is the best GaAs solar cell grown by HVPE at such a high growth rate. These results deserve to be published and are of interest to the III-V solar cell community. But beyond the III-V solar cell community there are little applications which benefit from such high GaAs growth rates. I am therefore not sure if this work is suitable for Nature Communications or should rather be published in a solar cell journal like IEEE JPV or Progress in Photovoltaics.

Thanks to the reviewer for the kind words. First, we would like to point out that the 29.1% result is grown using a different structure than the one presented in our work. That MOVPE-grown structure includes an AlInP window layer that decreases parasitic absorption. We are currently developing Al-containing materials in our reactor, and in the future, we hope to be able to grow the same higher efficiency structure that resulted in the record device. We believe that our current progress nonetheless validates that the inherent quality of HVPE-grown material is not an impediment to reaching the same heights as MOVPE-material.

Next, we respectfully disagree with the reviewer that there do not exist other applications that could benefit from ultra-high deposition rates. For example, Metaferia *et al.*,¹ and Barrios *et al.*,² demonstrated the use of high growth rate HVPE for the fabrication of buried heterostructure quantum cascade lasers and vertical cavity surface emitting lasers by growing semi-insulating InP and GaInP, respectively, in deep etched laser mesas. Those studies show that high growth rate is crucial not only for cost reasons but also to avoid active layer degradation due to simultaneous annealing. Lynch *et al.*,³ recently used HVPE to grow ~ 600-μm-thick orientation patterned GaAs for non-linear optical frequency conversion applications. In addition to these existing applications, we are hopeful that new applications yet to be developed will benefit from ultra-high growth rates. We also believe that photovoltaics are of general interest to the wide community. Thus, we believe the wider audience of Nature Communications is appropriate for these results.

¹Metaferia *et al.*, “Demonstration of a quick process to achieve buried heterostructure quantum cascade laser leading to high power and wall plug efficiency” *Opt. Eng.* 53(8) 087104

²Barrios *et al.*, “GaAs/AlGaAs buried-heterostructure vertical-cavity surface-emitting laser with semi-insulating GaInP:Fe regrowth”, *Electronics Letters*, Volume 36, Issue 18, August 2000, p. 1542 - 1544

³Lynch *et al.*, “Thick orientation-patterned GaAs grown by low-pressure HVPE on fusion-bonded templates”, *Journal of Crystal Growth* 353 (2012) 152–157

1. The manuscript is well organized, the length is appropriate to the content and the results well described. Still I believe that some improvements should be done:- discussion of Fig.2 refers to green data but the data shown is red.

We thank the reviewer for this, and we have changed the text in the manuscript.

2. I do not understand how a higher H₂ carrier gas flow which is injected together with the same amount of AsH₃ changes the amount of H radicals in the reactor. There is not more AsH₃ and why should this influence the hydrogen radical concentration in the reactor? I would recommend further experiments to clarify this influence of carrier gas flow on the growth rate of GaAs. Was the total H₂ flow through the HVPE machine kept constant in these experiments? What happens if the H₂ carrier gas is increased even further? Is there a

jetting inside the reactor? Is it possible that the AsH₃ distribution in the gas phase is changing? Maybe perform some modelling of the gas inside the reactor as a function of the carrier gas flow rates?

The key to enhancing the growth rate is delivering the hydride molecule to the growth front before it cracks to form As₂/As₄, a form through which the growth proceeds much more slowly. Delivery of the hydride molecule at high velocity, by increasing the carrier gas flow, limits the time it spends in the hot reactor and thus the probability that it will crack. This effect was studied and elucidated in Refs. 14 and 15, and so we do not believe that further experiments to study this effect are necessary. As was stated in the manuscript (both in the text and in the legend of Fig. 2), the total H₂ carrier flow through the reactor was kept constant (~10250 sccm), but we increased the growth rate by flowing proportionally more of the total carrier through the high velocity AsH₃ delivery tube. Based on our results shown in Fig. 2(b), we expect that increasing the carrier gas flow even more will further increase the growth rate, but we cannot do this experiment because our current H₂ carrier MFC was pushed to its limit in this study.

3. Another important finding of the paper in Figure 4 is that the GaAs EL2 trap density is more or less constant with growth rate and that this result differs from MOVPE growth reports where EL2 density increased significantly with growth rate. I believe that this result requires more discussion and maybe also additional experiments. It is unclear to the reviewer why there should be a difference between HVPE and MOVPE for the same V/III ratios? Both processes are gas phase processes and the EL2 should be a result of excess As on the surface which incorporates with higher probability as antisite if the growth rate is increased. I do not understand why HVPE would not show this effect. Therefore, the authors should discuss V/III ratio variation, influence of reactor pressure (?) and other possible influences on the observed EL2 density. V/III ratios should be given for the MOVPE and HVPE experiments.

While both growth techniques may be gas phase processes, the composition of the surface, the kinetics of growth and the energetics of adsorption and desorption are quite different in each case, and so it should not be assumed *a priori* that defect incorporation mechanisms are the same. As one example, in MOVPE, the growth rate shows almost no dependence on the V/III ratio or AsH₃ flow rate, whereas a correlation of growth rate and AsH₃ flow rate is *expected* in HVPE growth. In fact, we observe that effect in this study, implying that the extra As is being consumed at open As surface sites (rather than at Ga sites to form As_{Ga}). Furthermore, in the case where we increase the growth rate simply by increasing the H₂ carrier flow, we are not changing the balance of As/Ga in the reactor, or presumably at the surface, since equal amounts of As and Ga are consumed at the correct crystal sites.

We do not wish to speculate on the effects of reactor pressure, or other influences on EL2, since we did not vary these parameters in the study. We note that the V/III ratios were similar between our samples and those in Schmieder, *et al.*, and we noted in the manuscript that this implies that the EL2 incorporation mechanisms are likely very different between the growth methods.

4. Also Figure 4 shows some increase in EL2 density from 100 to 200 $\mu\text{m}/\text{h}$ and all HVPE results are above the low growth rate MOVPE material. I believe that this requires more discussion.

We are unclear what discussion is requested here. It is true that our V_{OC} values are higher than those presented for the MOVPE-grown devices, despite the high-growth-rate HVPE-grown cells having a higher measured concentration of EL2. There are many potential ways to degrade V_{OC}, with increased EL2 concentrations being one of those ways. However, in this case it is more likely that there are differences in the structures of the MOVPE and HVPE devices that play a more crucial role. In fact, the authors of the MOVPE study claimed that their solar cells are not limited only by EL2 defects. They showed that even though the defect density could be reduced by increasing the growth temperature, the V_{OC} of the cell grown at high temperature and high growth rate is still lower than the cell grown at lower temperature and lower growth rate, despite a lower EL2 concentration. The authors indicated that the V_{OC} of the high growth rate cell is limited by rear surface recombination. We believe that low EL2 concentrations are a necessary, but insufficient, means of producing high V_{OC}. Please also see our reply to Reviewer #1, point 11.

5. Some references did not display correctly on my computer and there is a Mm/h in Ref. (4) which should be changed to $\mu\text{m/h}$

We thank the reviewer for pointing out this, we corrected this in the manuscript.

Thank you again to the reviewers for their time and effort to carefully review our manuscript. We believe our revised manuscript is clearer and stronger after addressing their concerns, and hope that it is now suitable for publication in Nature Communications

Best Regards

Wondwosen Metaferia (Ph.D)
National Renewable Energy Laboratory

Reviewers' comments:

Reviewer #1 (Remarks to the Author):

The replies to reviewers helped to clarify and improve the manuscript.

I would still add a sentence in the conclusion about estimated GR values necessary to achieve the 1min total growth time and underline the sensitivity of this total time to GaInP GR.

Reviewer #2 (Remarks to the Author):

I think the authors properly did a great job of taking the reviewer comments and adding to the paper. I have some suggestions and questions that may further enhance the quality of the paper.

1. You mentioned that EL2 trap density is independence on the growth rate in HVPE. Did you also evaluate the minority carrier lifetime for n-GaAs base layer? I'm interested in the relationship between the carrier diffusion length and the base layer thickness.
2. In fig. 6, the cell performance seems to be slightly degraded with increasing the growth rate, though you mentioned "high material quality can be maintained at these extremely high growth rate" in abstract. What was the cause of these slight degradation? Otherwise, is there a window to improve the cell performance for especially cells with higher growth rate?
3. There is no suggestion on the growth regime for InGaP. Were InGaP films also grown under mass-transport regime in your D-HVPE, like GaAs?
4. In fig. 7(b) – (d), there is no comment on the requirement for the growth time analysis. Did you calculate the growth time in your D-HVPE system or future in-line, optimized HVPE system?

Reviewer #3 (Remarks to the Author):

Thank you for giving further explanations and for your response to my questions.

The authors have now added on page 3: "This means the use of a low pressure HVPE system allows access to the hydride enhanced mechanism to enhance the growth rate." I do not understand what the authors want to say with this statement? Very high growth rates and efficiencies are demonstrated by the authors at atmospheric pressure. Why would the growth rate increase further towards lower pressure? Is there evidence for this? Is there a limitation of pyrolysis at the growth temperature of 650°C. This paragraph should be written more clearly and as pointed out in my first review, it would be valuable to discuss the influence of reactor pressure on the results.

In my first review, I questioned the explanation why higher H₂ carrier gas flow in the AsH₃ line increases the growth rate. The authors respond that more uncracked AsH₃ can be delivered to the growth front if the H₂ flow is high and I can follow this explanation but it remains doubtful to me that this would result in a higher growth rate. AsH₃ has to decompose to release As at least at the growth interface. I do not see clear evidence in the data presented for the hypothesis that As₂/As₄ is retarding the growth. I believe that this is still a shortcoming and that the explanation given on page 7-8 is not sufficiently supported by the experimental data. Modelling of decomposition in the reactor would help to understand the data and motivate the results.

Page 8 please correct spelling error: "reoptimization"

In my first review, I also expressed concerns about the discussion of the EL2 defect incorporation in MOVPE versus HVPE grown samples. The authors did not make changes to the manuscript except for mentioning that similar V/III ratios were used in their HVPE experiments and in MOVPE results from literature. I believe that this is not sufficient and that further discussion and maybe experiments are required to understand the differences between MOVPE and HVPE. Also it has to be understood if there are differences in the determination of the EL2 concentration. The authors used the rear-heterojunction solar cell for the DLTS measurement. The DLTS signal should be mainly resulting from the lower doped GaAs layer but what is the influence of the interface between GaInP and GaAs which is in the middle of the space charge region? Was the MOVPE EL2 density also determined from a rear-heterostructure GaAs solar cell? The authors are not giving any reason for the lower EL2 density in the HVPE grown GaAs compared to MOVPE and I believe that more understanding and validation is necessary before publication of this result. Also it would be helpful to include error bars to the EL2 defect density.

The authors responded to reviewer #3(3) about AsH₃ effect on growth rate and H₂ flow rate but I do not understand what they want to say. They state that the H₂ flow rate with the AsH₃ does not influence the As/Ga ratio. But before, they showed that it increases the growth rate with the argument that more AsH₃ is brought to the growth surface. Therefore, the chemical composition at the growth interface is significantly altered. Otherwise one could not explain the result. I still have the opinion that this EL2 discussion should not be published without further evidence and discussion.

In the response to reviewer #1(11) the authors state that the devices are most likely not limited by EL2, in response to reviewer #3(4) they say that the MOVPE grown solar cells were most likely also not limited by the EL2 defect but by rear surface recombination. If this is the case, the EL2 section could be completely removed from the paper. It is possible, that the MOVPE grown solar cell results were significantly influenced by interfaces and that the switching of gases during changes from GaAs to GaInP is more critical to the device performance than the EL2 concentration. But this discussion seems to be out of the scope of the presented work.

Further comments: I believe that the influence of the contact layer doping on the ohmic resistance is a minor detail and could be removed just stating the desired doping level.

I believe that the section "Device growth time at high growth rates" is interesting but it is confusing that GaInP/GaAs tandem solar cells are discussed which have not been the subject of previous chapters. In fact the GaInP which was used in the GaAs solar cells was grown at rates of 2.3 μm/h for the etch stop and window and 6 μm/h for the BSF layers as mentioned in the beginning of the manuscript. I believe that therefore only the data on GaAs solar cells should be discussed in the last section.

Reviewer #1 (Remarks to the Author):

The replies to reviewers helped to clarify and improve the manuscript.

1. I would still add a sentence in the conclusion about estimated GR values necessary to achieve the 1min total growth time and underline the sensitivity of this total time to GaInP GR.

We agree with the reviewer that it is a good idea to include a sentence in the conclusion about the GaInP growth rate and total growth time. We made the corresponding change in the manuscript.

Reviewer #2 (Remarks to the Author):

I think the authors properly did a great job of taking the reviewer comments and adding to the paper. I have some suggestions and questions that may further enhance the quality of the paper.

1. You mentioned that EL2 trap density is independence on the growth rate in HVPE. Did you also evaluate the minority carrier lifetime for n-GaAs base layer? I'm interested in the relationship between the carrier diffusion length and the base layer thickness.

We did not evaluate the minority carrier lifetime for n-GaAs base layer. However, it is interesting to study the relationship between carrier diffusion length and growth rate and we will consider doing this in the future. We would like note here that, as can be seen from the QE plot in the manuscript (Figure 6 (d)), we don't see any change in carrier collection with growth rate. Therefore we can at least say, if the diffusion length changed it never got low enough to affect the carrier collection in our cells.

2. In fig. 6, the cell performance seems to be slightly degraded with increasing the growth rate, though you mentioned "high material quality can be maintained at these extremely high growth rate" in abstract. What was the cause of these slight degradation? Otherwise, is there a window to improve the cell performance for especially cells with higher growth rate?

We haven't done as many optimizations at higher growth rates (>200 $\mu\text{m}/\text{h}$) as we did for lower growth rates. As noted in the text, we have used the same recipe for higher growth rate cells as the lower growth rate cells, which affected, for example, the doping density in different device layers. Further optimizations with doping of each of the layers would improve the cell performance.

3. *There is no suggestion on the growth regime for InGaP. Were InGaP films also grown under mass-transport regime in your D-HVPE, like GaAs?*

We don't have enough data to identify the growth window for the GaInP growth at this time. The GaInP was grown at 650°C and varying partial pressures of GaCl and InCl. It is likely that the growth window is mass transport limited, the same as for our GaAs. However, we don't want to speculate about the growth window because we don't have supporting data.

4. *In fig. 7(b) – (d), there is no comment on the requirement for the growth time analysis. Did you calculate the growth time in your D-HVPE system or future in-line, optimized HVPE system?*

All the analyses for growth time were done by combining achieved growth rate results of GaAs and GaInP by our current D-HVPE. The analyses assume a future in-line, optimized HVPE system. We stated this in the manuscript and we have now added a sentence in the caption to address this.

Reviewer #3 (Remarks to the Author):

Thank you for giving further explanations and for your response to my questions.

1. *The authors have now added on page 3: “This means the use of a low pressure HVPE system allows access to the hydride enhanced mechanism to enhance the growth rate.” I do not understand what the authors want to say with this statement? Very high growth rates and efficiencies are demonstrated by the authors at atmospheric pressure. Why would the growth rate increase further towards lower pressure? Is there evidence for this? Is there a limitation of pyrolysis at the growth temperature of 650°C. This paragraph should be written more clearly and as pointed out in my first review, it would be valuable to discuss the influence of reactor pressure on the results.*

We thank the reviewer for pointing out this issue. In the original manuscript, we brought up the concept of hydride-enhanced epitaxy without explaining what it was until later in the growth optimization section. We have now included an explanation of this mechanism in this earlier paragraph and cited the references appropriately.

As we explained in the manuscript and in our previous reply to the reviewer's comment, the key in the hydride growth mechanism is the delivery of uncracked AsH₃ at the growth site. Please refer the work of K. L. Schulte et al., Appl. Phys. Lett. 112, 042101 (2018) and K. Grüter et al., J. Cryst. Growth 94(3), 607 (1989). Specifically, the work of Grüter et al., clearly detailed the hydride enhanced growth mechanism by a low-pressure system (~8 Torr) and explained the advantage of low pressure in keeping AsH₃ uncracked before reaching the growth site and how that enhanced the growth rate. The dependence of reactor pressure on growth rate is also clearly explained. We, therefore, believe that citing the reference should suffice and that it would be inappropriate to reproduce published data that are beyond the scope of the current work.

2. *In my first review, I questioned the explanation why higher H₂ carrier gas flow in the AsH₃ line increases the growth rate. The authors respond that more uncracked AsH₃ can be delivered to the growth front if the H₂ flow is high and I can follow this explanation but it remains doubtful to me that this would result in a higher growth rate. AsH₃ has to decompose to release As at least at the growth interface. I do not see clear evidence in the data presented for the hypothesis that As₂/As₄ is retarding the growth. I believe that this is still a shortcoming and that the explanation given on page 7-8 is not sufficiently supported by the experimental data. Modelling of decomposition in the reactor would help to understand the data and motivate the results.*

In a hot wall reactor, AsH₃ rapidly decomposes to form As₂ and As₄ at temperatures above ~ 400 °C, well below typical growth temperatures, (SP DenBaars, et al., Journal of Crystal Growth 77 (1-3), 188 (1986) and Vladimir S. Ban et al., Journal of Crystal Growth 17, 19 (1972)). In these cases, the reduction of AsGaCl surface complexes by hydrogen to form GaAs and HCl is the rate-limiting step, resulting in the high kinetic barrier (~200 kJ/mol) to growth and reduces the growth rate. Therefore, high temperature is needed in order to reduce this kinetic barrier and increase the growth rate. However, with uncracked AsH₃, the growth reaction at the surface directly involves the AsH₃ molecule ($\text{GaCl} + \text{AsH}_3 \rightleftharpoons \text{GaAs} + \text{HCl} + \text{H}_2$) which does not require the presence of H₂, and the kinetic barrier for such a reaction was found to be very low (~9 kJ/mol) (K. L. Schulte et al., Appl. Phys. Lett. 112, 042101 (2018)). That reference, which is cited and briefly summarized in the manuscript, provides direct evidence to support the enhancement of growth rate by uncracked AsH₃. Please also refer the reply to #1 above and the corresponding change we made in the manuscript.

3. Page 8 please correct spelling error: “reoptimization”

We thank the reviewer for catching this, and we have corrected this typo.

4. In my first review, I also expressed concerns about the discussion of the EL2 defect incorporation in MOVPE versus HVPE grown samples. The authors did not make changes to the manuscript except for mentioning that similar V/III ratios were used in their HVPE experiments and in MOVPE results from literature. I believe that this is not sufficient and that further discussion and maybe experiments are required to understand the differences between MOVPE and HVPE. Also, it has to be understood if there are differences in the determination of the EL2 concentration. The authors used the rear-heterojunction solar cell for the DLTS measurement. The DLTS signal should be mainly resulting from the lower doped GaAs layer but what is the influence of the interface between GaInP and GaAs which is in the middle of the space charge region? Was the MOVPE EL2 density also determined from a rear-heterostructure GaAs solar cell? The authors are not giving any reason for the lower EL2 density in the HVPE grown GaAs compared to MOVPE and I believe that more understanding and validation is necessary before publication of this result. Also, it would be helpful to include error bars to the EL2 defect density.

It is not clear to us that the type of device used to measure EL2 is important. The DLTS technique measures the perturbations to junction capacitance due to the filling and unfilling of trap states at the edge of the depletion region. The technique works for different types of junctions, including semiconductor/semiconductor junctions as well as semiconductor/metal junctions, as long as conditions exist in which the depletion approximation holds, as they do in the device design examined in this work. At no time during the measurement is the junction itself probed. The reviewer seems to wish for a concrete answer as to why MOVPE and HVPE-grown GaAs solar cells exhibit different levels of EL2. We would like to reiterate that understanding the EL2 defect incorporation mechanisms in MOVPE vs HVPE is not the scope of the present study. Our intention is to show the effect of growth rate on material quality. As much as we would like to understand the incorporation mechanisms, it is enough for the present work that the difference exists. Just because the atomic-level mechanisms leading to the observed trends are not fully elucidated does not mean that the data are invalid or unworthy of publication. We note that the original motivation for looking at EL2 concentrations in HVPE-grown material was a previous report of high-growth-rate solar cells grown by MOVPE that noted a drastic increase in EL2 with increasing growth rate. Our motivation was clearly stated in the manuscript.

5. The authors responded to reviewer #3(3) about AsH₃ effect on growth rate and H₂ flow rate but I do not understand what they want to say. They state that the H₂ flow rate with the AsH₃ does not influence the As/Ga ratio. But before, they showed that it increases the growth rate with the argument that more AsH₃ is brought to the growth surface. Therefore, the chemical composition at the growth interface is significantly altered. Otherwise one could not explain the result. I still have the opinion that this EL2 discussion should not be published without further evidence and discussion.

We do not see a contradiction here. The increase in growth rate is not due to an increase in the amount of arsenic brought to the surface, but rather the form of the arsenic. As stated several times before, increasing the hydrogen flow increases the ratio of AsH_3 to As_x species. AsH_3 has a much lower kinetic barrier to growth, thereby allowing increased growth rates, while the incorporation of As_x species is slowed by a larger kinetic barrier, presumably allowing time for excess As_2 and As_4 to desorb from the surface.

6. In the response to reviewer #1(11) the authors state that the devices are most likely not limited by EL2, in response to reviewer #3(4) they say that the MOVPE grown solar cells were most likely also not limited by the EL2 defect but by rear surface recombination. If this is the case, the EL2 section could be completely removed from the paper. It is possible, that the MOVPE grown solar cell results were significantly influenced by interfaces and that the switching of gases during changes from GaAs to GaInP is more critical to the device performance than the EL2 concentration. But this discussion seems to be out of the scope of the presented work.

We have already discussed our motivation for studying EL2 in our solar cells above. Indeed, the reviewer's apparent surprise that the EL2 densities could be different between the two growth techniques seems to be justification enough for inclusion of these data in our manuscript. It was not known *a priori* that EL2 will not limit solar cells at high growth rates, therefore it is valuable to present this study showing cells are not limited by an EL2 defect. The EL2 section also shows the insensitivity of trap density as growth rate increases. Therefore, we respectfully disagree with the reviewer and we would like to keep the EL2 section as it shows the general trend in the EL2 defect density with respect to the growth rate, which would otherwise remain a question in a reader's mind (i.e. What happens to the defect density as the growth rate increases?).

7. Further comments: I believe that the influence of the contact layer doping on the ohmic resistance is a minor detail and could be removed just stating the desired doping level.

We respectfully disagree with the reviewer and believe the influence of the growth rate on doping concentration should be discussed. In the context of the paper, it is very interesting to discuss doping at very high growth rate. To our knowledge, no one has ever demonstrated the impact of growth at rates of this magnitude on doping. Therefore, we believe it is interesting to the general audience of Nature Communications to understand all aspects of the device growth at these novel rates.

8. I believe that the section "Device growth time at high growth rates" is interesting but it is confusing that GaInP/GaAs tandem solar cells are discussed which have not been the subject of previous chapters. In fact, the GaInP which was used in the GaAs solar cells was grown at rates of $2.3\mu\text{m/h}$ for the etch stop and window and $6\mu\text{m/h}$ for the BSF layers as mentioned in the beginning of the manuscript. I believe that therefore only the data on GaAs solar cells should be discussed in the last section.

The entire discussion of GaInP growth rate was inspired by previous insightful questions from Reviewer #1 and #2. We have included GaInP growth rates up to $\sim 206\mu\text{m/h}$ and discussed the effect of GaInP growth rate on the total growth time for GaAs single junction solar cells. We believe doing the same analysis for the GaInP/GaAs tandem solar cell provides useful insight to the potential advantage of high growth rates of both GaAs and GaInP for increased throughput of multijunction solar cells and, crucially, other devices that require thick GaInP layers. We believe this is also interesting to the wide readership of Nature Communications.

Thank you again to the reviewers for their time and effort to carefully review our manuscript. We believe our revised manuscript is clearer and stronger after addressing their concerns, and hope that it is now suitable for publication in Nature Communications.

Best Regards

Wondwosen Metaferia (Ph.D)
National Renewable Energy Laboratory

Reviewers' comments:

Reviewer #2 (Remarks to the Author):

Thank you for responding properly for our suggestions and comments. I have some more comments as below that may clarify your discussions.

1. Regarding EL2 trap density evaluation, one of the possible options is removing MOVPE reference data from Fig. 4 and just stating the trap density was independent on the growth rate in D-HVPE not to expand the scope of your article. I agree with your discussion that the EL2 trap density characterization in Fig.4 strongly supports after-mentioned I-V curves measured for GaAs solar cells grown with various growth rates.
2. In solar cell performance, I think the performance may be determined by not only EL2 trap density but also heterointerface quality. Did you compare the abruptness of heterointerfaces between high and low growth rate by structural characterization like SIMS? Please comment on the effect of the growth rate on the abruptness of heterointerfaces, because high growth rates may result in broadened heterointerfaces.

Reviewer #3 (Remarks to the Author):

I am not satisfied with the revisions made by the authors. The authors should ensure that results are presented clearly with sufficient level of detail to understand the assumptions and methods being used as the base for the conclusions. Further your data must be analyzed thoroughly to avoid any source of misinterpretation. I have still significant concerns in both cases. Also I find that switching between reactor configurations, growth conditions and solar cell structures in different sections of the paper is misleading and can easily lead to confusion.

Reviewer #2(4) asked for the requirements for the growth time analysis in Fig. 7. The response of the authors is to add the following to the figure caption: "(c) total growth time for a GaInP/GaAs two-junction solar cell structure by a future optimized in-line HVPE reactor". What is this future optimized in-line HVPE reactor? There is no reference as far as I can see and no description of the reactor in the text. It is not acceptable that growth times for solar cell structures are discussed without any possibility for the reviewer and reader to verify if the assumptions are trustful. Heating, cooling, growth rate of each layer, switching between layers, transfer between chambers would be the minimum necessary to understand the assumptions. But as pointed out already in my last review, I disagree that the more complex GaInP/GaAs tandem solar cell structure which is not the subject of this paper is discussed only in the "Device growth time" section as there is not sufficient space to add all necessary details of this cell structure, the reactor and the assumptions made in the calculation. I do not see any value in discussing growth times in a future generation HVPE reactor which is not clearly defined and has never been built and which may never work. On the other hand, just multiplying growth rate with layer thickness and adding this up for a layer stack is something that should not fill pages of a review journal article. The result section of this publication should focus on what has been achieved and what can be concluded from it. With this being said, I still agree to motivate that with the high growth rates demonstrated, single-junction GaAs solar cells as discussed in the first part of the paper could be grown in xxx minutes if a reactor is assumed with heating time, cooling time, temperatures, ... for all layers, switching time between x and y, etc.. I guess this could be said in 2-3 sentences.

From my point of view, the paper has parts which are not well connected. Especially the new section "Device growth time at high growth rates" discusses different growth conditions, tandem cells and

assumes a different future reactor compared to the beginning of the paper. This is confusing. Example: In the Abstract you are saying: "We also show growth rates for GaInP using D-HVPE up to 206 $\mu\text{m}/\text{h}$, without signs of saturation under the tested conditions." But then in the experimental part you are referring to "GaInP layers were grown at 2.3 $\mu\text{m}/\text{h}$ for the etch stop and window and 6 $\mu\text{m}/\text{h}$ for the BSF". My understanding is that these are the growth rate which have been used for all the experimental GaAs solar cells. It is very misleading to cite a higher growth rate in the abstract, then use low growth rates in the experimental section and then again show some higher GaInP growth rates in the "Device growth time" section but with the addition that material quality is unknown. Readers are quickly misled and I believe that the paper must be significantly restructured and focused on specific results before being published.

I have pointed out several times that the discussion of the EL2 defect in GaAs grown at different rates is not sufficiently supported by the data presented in the manuscript and that there is no interpretation which would suggest such a large difference between MOVPE and HVPE. The authors reply that my "apparent surprise" would be enough justification to publish this result. But I respectfully disagree! If the result is not expected then we should first question if the measurement and interpretation of the data is correct! And this is not done with sufficient care. Is the comparison with Ref. 16 possible? Schmieder et al have used pn-junctions in GaAs and not a rear heterocell. I had asked for error bars and the authors did not respond to this. The graph in Fig. 4 remains unchanged. DLTS is not strait forward and results can clearly differ when the method is applied to different samples and under different conditions. Usually, DLTS is done on Schottky junctions where the space charge region only extends into the semiconductor. Here DLTS was performed on a rear-heterojunction solar cell with a 30 nm GaInP p-layer. The DLTS signal results from the edge of the space charge region as the authors are pointing out in the response letter and this is naturally in the p and n-type layer! It is probably assumed that the p-type GaInP is highly doped and therefore does not significantly contribute to the DLTS signal. But the GaInP is only 30 nm thin and may be strongly influenced by surface states (Fermi level pinning). Is it then still valid to ignore the influence of the GaInP on the DLTS result? Also I am still not sure if the heterointerface between GaInP and GaAs could have an influence on the measurement result? Further, how accurately was the doping of the n-GaAs base layer determined? Was it determined for each sample and the value used for the calculation of the EL2 defect density from the DLTS signal? Did you vary the pulse width and applied voltage during the DLTS measurement to see if this has an influence on the charging / discharging of the defect states? Did you process Schottky junctions to compare the results? What are the exact conditions during the DLTS measurement? I keep my opinion that the EL2 results need further verification and discussion before they can be published and they could be a paper on its own if the differences between MOVPE and HVPE are real and understood.

And I am also puzzled that the authors have included only part of the data from reference 16 which shows high levels of EL2 in GaAs. From my point of view this is unacceptable as Schmieder et al. also showed one data point for a high growth rate sample which had comparable EL2 density as the HVPE data presented by the authors. The authors mention in the text that the EL2 density was an order of magnitude lower for higher growth temperatures in Ref. 16 but the data point should also be added to Fig. 4 and then the question is: What should we learn? Maybe the EL2 density is a function of temperature?

Additionally:

- Growth rate discussion – it would be helpful to add absolute values for the efficiency of Ga and AsH₃ utilization in these experiments. This is essential for the cost of the overall process.
- I keep my opinion that the discussion of cap layer doping control should be removed and is a minor technical detail, not suitable for a Nature Communication article

In response to my earlier remark 1 and 2, I have been reading the two references of K. Schulte in App. Phys. Letters and K. Grüter in J. Cryst. Growth and agree that these papers discuss the influence of HVPE reactor pressure on growth rate sufficiently.

Reviewers' comments:

Reviewer #2 (Remarks to the Author):

Thank you for responding properly for our suggestions and comments. I have some more comments as below that may clarify your discussions.

1. Regarding EL2 trap density evaluation, one of the possible options is removing MOVPE reference data from Fig. 4 and just stating the trap density was independent on the growth rate in D-HVPE not to expand the scope of your article. I agree with your discussion that the EL2 trap density characterization in Fig.4 strongly supports after-mentioned I-V curves measured for GaAs solar cells grown with various growth rates.

We previously included the MOVPE DLTS data along with our own because it provides motivation for studying the same defect in our material as a function of growth rate. That is, others saw a change in defect density with growth rate – will we see the same effect? There was never meant to be a direct comparison between the data from the two different growth methods. We now realize, however, that a reader may be misled by the inclusion of the MOVPE data, believing, perhaps, that we are directly comparing material quality between HVPE and MOVPE. As such, we now believe that removing the MOVPE data is reasonable and have removed it from that figure to allow our data to stand on its own. We do still use the existence of the MOVPE data (referenced in our manuscript) as motivation for the study of the EL2 defect in the HVPE-grown material.

2. In solar cell performance, I think the performance may be determined by not only EL2 trap density but also heterointerface quality. Did you compare the abruptness of heterointerfaces between high and low growth rate by structural characterization like SIMS? Please comment on the effect of the growth rate on the abruptness of heterointerfaces, because high growth rates may result in broadened heterointerfaces.

We did not characterize this directly, but the QE data in Fig. 4 indicate that interfacial recombination has not increased for the higher growth rate devices. The rear heterojunction device is extremely sensitive to surface recombination at the front interface, but we do not observe any changes in the short wavelength QE indicative of interface degradation.

Reviewer #3 (Remarks to the Author):

I am not satisfied with the revisions made by the authors. The authors should ensure that results are presented clearly with sufficient level of detail to understand the assumptions and methods being used as the base for the conclusions. Further your data must be analyzed thoroughly to avoid any source of misinterpretation. I have still significant concerns in both cases. Also I find that switching between reactor configurations, growth conditions and solar cell structures in different sections of the paper is misleading and can easily lead to confusion.

Reviewer #2(4) asked for the requirements for the growth time analysis in Fig. 7. The response of the authors is to add the following to the figure caption: “(c) total growth time for a GaInP/GaAs two-junction solar cell structure by a future optimized in-line HVPE reactor”. What is this future optimized in-line HVPE reactor? There is no reference as far as I can see and no description of the reactor in the text. It is not acceptable that growth times for solar cell structures are discussed without any possibility for the reviewer and reader to verify if the assumptions are trustful. Heating, cooling, growth rate of each layer, switching between layers, transfer between chambers would be the minimum necessary to understand the assumptions. But as pointed out already in my last review, I disagree that the more complex GaInP/GaAs tandem solar cell structure which is not the subject of this paper is discussed only in the “Device growth time” section as there is not sufficient space to add all necessary details of this cell structure, the reactor and the assumptions made in the calculation. I do not see any value in discussing growth times in a future generation HVPE reactor which is not clearly defined and has never been built and which may never work. On the other hand, just multiplying growth rate with layer thickness and adding this up for a layer stack is something that should not fill pages of a review journal article. The result section of this publication should focus on what has been achieved and what can be concluded from it. With this being said, I still agree to motivate that with the high growth rates demonstrated, single-junction GaAs solar cells as discussed in the first part of the paper could be grown in xxx minutes if a reactor is assumed with heating time, cooling time, temperatures, ... for all layers, switching time between x and y, etc.. I guess this could be said in 2-3 sentences.

We thank the reviewer for their in-depth reading of this section. We first point out that other reviewers asked for this analysis after the first review, and this section was added to address those concerns. Thus, we believe that there is value in the analysis, but agree that there were weaknesses in the presentation, and have significantly edited the section. We have moved the GaInP growth data to the earlier section about achieving high GaAs growth rate where it fits more logically and added associated discussion that compare and contrast the pure growth results in that section. In the analysis of growth times, we have now described the in-line reactor assumed more completely in the text, added caveats for the reader to consider, and have included a reference that contains an illustration of the reactor in order to remove any sources of confusion. The section now begins:

“In this section, we calculated the influence of growth rate on the total growth time of a single-junction GaAs solar cell, with the structure described in Fig. 1(b), to provide insight into the benefit of high growth rates on overall device throughput. Here, we assume the modular in-line reactor described in ref. ⁶ in which each layer is deposited in a separate chamber as the wafer is shuttled through. Such a reactor would not have constraints that limit the growth rate of specific layers as in our dual chamber reactor, and the in-line nature also minimizes the impact of heat-up and cool-down times. In an in-line system, each deposition zone would always be in steady state, and interfacial abruptness is controlled by the shape of the gas curtains.³¹ This reactor does not yet exist, but the in-line concept is best able to make use of these high growth rates, and we wish to show growth times that may ultimately be achievable. We recognize that this design is perhaps aspirational, but similar in-line tools are in use for the deposition of CdTe solar cells.”

We respectfully disagree that it is not appropriate to discuss the growth of a dual-junction device in this section. These are common III-V devices, and furthermore, we have demonstrated their growth by D-HVPE in a previous reference that is cited in this manuscript. We now note explicitly in the present manuscript that the main difference between the two-junction structure and the present structure is the ~1 μm thick GaInP

absorber layer, which contributes most to the overall growth time. We hope that this additional information helps to clarify the results of this section.

From my point of view, the paper has parts which are not well connected. Especially the new section “Device growth time at high growth rates” discusses different growth conditions, tandem cells and assumes a different future reactor compared to the beginning of the paper. This is confusing. Example: In the Abstract you are saying: “We also show growth rates for GaInP using D-HVPE up to 206 $\mu\text{m}/\text{h}$, without signs of saturation under the tested conditions.” But then in the experimental part you are referring to “GaInP layers were grown at 2.3 $\mu\text{m}/\text{h}$ for the etch stop and window and 6 $\mu\text{m}/\text{h}$ for the BSF”. My understanding is that these are the growth rate which have been used for all the experimental GaAs solar cells. It is very misleading to cite a higher growth rate in the abstract, then use low growth rates in the experimental section and then again show some higher GaInP growth rates in the “Device growth time” section but with the addition that material quality is unknown.

Readers are quickly misled and I believe that the paper must be significantly restructured and focused on specific results before being published.

We have restructured the manuscript and we believe, and hope that the reviewer agrees, that what was accomplished in this work is now clearer to the reader. All of the growth rate data for individual layers is contained in one section, where we only discuss the growth results for GaAs and GaInP epilayers, not devices. We’ve also updated the experimental section to explicitly state what growth rates were used in the devices, and what our motivation was. This section now reads:

“GaInP layers were grown at 2.3 $\mu\text{m}/\text{h}$ for the etch stop and window and 6 $\mu\text{m}/\text{h}$ for the BSF layers and were the same for all solar cells in this study. The growth conditions for the GaInP layers in the solar cell structures were kept constant in order to relate changes in device performance to changes in the GaAs material quality as much as possible. Furthermore, these relatively low GaInP growth rates are necessary in the cell structures to provide sufficient time to accommodate the gas flow changes needed for the different device layers due to the constraint of only two reactor chambers.²⁰ Thus, we did not incorporate the high growth rate GaInP into these devices in this work. We note that we previously incorporated 54 $\mu\text{m}/\text{h}$ GaInP into two-junction GaInP/GaAs solar cells that showed $V_{\text{OC}} = 1.41$ V despite an passivated structure.¹²”

We disagree that the GaInP material quality is “unknown”. As we have cited in the manuscript, the GaInP used in the tandems in reference 12 was grown at 54 $\mu\text{m}/\text{h}$, and exhibited a $V_{\text{OC}} = 1.41$ V, quite respectable for a device without a window layer. In the present manuscript, we have demonstrated GaAs with very little decrease in quality up to 300 $\mu\text{m}/\text{h}$, generally suggesting that HVPE material quality does not degrade significantly with growth rate. While nothing can replace demonstration of a device at the highest GaInP growth rate, we do not believe it is unreasonable to use the highest demonstrated rate in the throughput analysis. We also believe that this will no longer be misleading to the reader now that our goals and motivation are stated explicitly up front.

I have pointed out several times that the discussion of the EL2 defect in GaAs grown at different rates is not sufficiently supported by the data presented in the manuscript and that there is no interpretation which would suggest such a large difference between MOVPE and HVPE. The authors reply that my “apparent surprise” would be enough justification to publish this result. But I respectfully disagree! If the result is not expected then we should first question if the measurement and interpretation of the data is correct! And this is not done with sufficient care. Is the comparison with Ref. 16 possible? Schmieder et al have used pn-junctions in GaAs and not a rear heterocell. I had asked for error bars and the authors did not respond to this. The graph in Fig. 4 remains unchanged. DLTS is not strait forward and results can clearly differ when the method is applied to different samples and under different conditions. Usually, DLTS is done on Schottky junctions where the spacecharge region only extends into the semiconductor. Here DLTS was performed on a rear-heterojunction solar cell with a 30 nm GaInP p-layer. The DLTS signal results from the edge

of the space charge region as the authors are pointing out in the response letter and this is naturally in the p and n-type layer! It is probably assumed that the p-type GaInP is highly doped and therefore does not significantly contribute to the DLTS signal. But the GaInP is only 30 nm thin and may be strongly influenced by surface states (Fermi level pinning). Is it then still valid to ignore the influence of the GaInP on the DLTS result? Also I am still not sure if the heterointerface between GaInP and GaAs could have an influence on the measurement result? Further, how accurately was the doping of the n-GaAs base layer determined? Was it determined for each sample and the value used for the calculation of the EL2 defect density from the DLTS signal? Did you vary the pulse width and applied voltage during the DLTS measurement to see if this has an influence on the charging / decharging of the defect states? Did you process Schottky junctions to compare the results? What are the exact conditions during the DLTS measurement? I keep my opinion that the EL2 results need further verification and discussion before they can be published and they could be a paper on its own if the differences between MOVPE and HVPE are real and understood.

And I am also puzzled that the authors have included only part of the data from reference 16 which shows high levels of EL2 in GaAs. From my point of view this is unacceptable as Schmieder et al. also showed one data point for a high growth rate sample which had comparable EL2 density as the HVPE data presented by the authors. The authors mention in the text that the EL2 density was an order of magnitude lower for higher growth temperatures in Ref. 16 but the data point should also be added to Fig. 4 and then the question is: What should we learn? Maybe the EL2 density is a function of temperature?

We thank the reviewer for this thorough explanation of their view of this section of the manuscript. We believe that we now understand the concerns well enough to be able to respond appropriately. First, a quick clarification of a misconception. The GaInP p-layer is 300 nm thick, as shown in Fig. 1(b), not 30 nm. We believe that the thickness of this layer, combined with the relatively high doping level, will eliminate any effects from carrier depletion due to Fermi level pinning at the surface.

Next, as discussed above in the response to Reviewer #2, we also now see how inclusion of the MOVPE DLTS data could potentially mislead a reader into thinking that we are directly comparing defect levels, rather than using the previous report as motivation for our study. Therefore, we have removed the MOVPE data from Fig. 5.

If we understand correctly, the main point of contention is that we presented defect level data from two different, albeit related, growth techniques that disagreed in their trend with increasing growth rate. That is, we showed EL2 defect density in HVPE-grown material that was much less sensitive to growth rate than previous reports. Again, if we understand correctly, the reviewer would like to see this discrepancy explained, or at least have some guarantee that the data were collected and analyzed properly. Of course, we agree with the reviewer wholeheartedly. To the first point, understanding the difference in EL2 defect levels between HVPE and MOVPE, we agree with the reviewer that this would indeed be a paper unto itself (or perhaps a Ph.D. thesis or two)! We would like to point out that there is no reason to expect the incorporation of EL2 to be the same in HVPE and MOVPE, despite both of them being vapor phase growth techniques. There are numerous differences between the two techniques that could lead to different trends in EL2 incorporation. For one example, the fact that the HVPE growth rate is demonstrably a function of group V overpressure, in complete contrast to MOVPE, illustrates that the role of As in the growth and on the surface is not the same as in MOVPE growth. In addition, Schulte and Kuech (<https://aip.scitation.org/doi/abs/10.1063/1.490>) show that EL2 dependence in HVPE GaAs is a function of thermodynamics, kinetics, and surface composition, and not a pure function of growth rate as has been observed in MBE and MOVPE. The data in the present manuscript are acquired from materials grown in a different growth regime entirely (the hydride-enhanced regime), so it is difficult to assume the dependence of EL2 concentration on growth rate *a priori*.

As stated above, we agree that the differences in EL2 concentrations are important to understand, but we maintain that this level of detail is beyond the scope of the current work and is not necessary to understand our conclusions. We conclude in this work that the effects of growth rate on our solar cell performance are small-to-negligible, as borne out by our measurements of the open-circuit voltage (V_{OC}). The measurements

of EL2 levels provide support for this conclusion. If there were as drastic an increase in EL2 concentration as observed in previous reports, it is impossible that these high V_{OC} values could be obtained.

As further support for the role of EL2 in our devices, we were recently able to obtain a pulse generator capable of the short pulses necessary to directly measure the capture cross section of this defect. We have now measured the capacitance transient behavior as a function of pulse width with the following results:

$N_D = 2.2 \times 10^{16} \text{ cm}^{-3}$ (from C-V)
 $N_t = 3.0 \times 10^{14} \text{ cm}^{-3}$ (from DLTS)
 $\sigma_\infty = 1.3 \times 10^{-13} \text{ cm}^2$ (from DLTS)

The slope from the plot is the capture rate, $-c_n$ in ns^{-1} .

$$c_n = \sigma v_{th} N_t \quad v_{th} = \sqrt{\frac{3k_B T}{m^*}}$$

where $m^* = 0.067m_0$

For $c_n = 4.5 \times 10^6 \text{ s}^{-1}$ at $T=345\text{K}$,
 $\sigma = 3.1 \times 10^{-16} \text{ cm}^2$
 $\tau = 1/c_n = 220 \text{ ns}$

For $c_n = 5.8 \times 10^6 \text{ s}^{-1}$ at $T=373\text{K}$,
 $\sigma = 3.8 \times 10^{-16} \text{ cm}^2$
 $\tau = 1/c_n = 170 \text{ ns}$

This more direct measurement of the capture cross section allows us to calculate a lifetime for the EL2 defect in our materials, which was one of the earlier requests from the reviewer. The non-radiative lifetime is longer than the radiative lifetime expected in GaAs for the measured doping levels, indicating that it is not fully responsible for limiting device performance. However, it is within a factor of two, which implies that it could be playing a role. The slight decrease in V_{OC} for the highest measured growth rates could plausibly be related to the increased EL2 concentrations.

To the second point, showing that the data are trustworthy, we point out that the measured electron capture cross section that we obtain is $3\text{-}4 \times 10^{-16} \text{ cm}^2$, which is the same value others report in the literature for EL2 (see new refs 28 and 29 in the manuscript). The activation energy of the trap was 0.82 eV, exactly what one would expect for EL2 in GaAs. There is also only one defect apparent in the raw DLTS data (now shown in the Supplemental Information), making analysis straightforward. All of our data indicate that the defect is behaving exactly as EL2 in GaAs should. This gives us great confidence that there is nothing wrong with either the data collection methods or the analysis. We further point out that we have significant experience with the DLTS method, including previous work to explain physical effects that had been observed in the literature but that were not at all understood (S. W. Johnston, S. R. Kurtz, D. J. Friedman, A. J. Ptak, R. K. Ahrenkiel, and R. S. Crandall, "Observed trapping of minority-carrier electrons in p-type GaAsN during deep-level transient spectroscopy measurement," *Appl. Phys. Lett.* 86, 072109 (2005) and S. Kurtz, S. Johnston, and H. M. Branz, "Capacitance-spectroscopy identification of a key defect in N-degraded GaInNAs solar cells," *Appl. Phys. Lett.* 86, 113506 (2005)).

Addressing other concerns of the reviewer:

The difference in doping density is more than two orders of magnitude between the n and p sides of the junction. We believe that the assumption that the modulation of the depletion region occurs nearly entirely in

the GaAs layer is a good one. We cannot completely discount the possibility that the GaInP plays some role in the measurement, but we point to the measured activation energy, capture cross section, and single-defect nature of the data to suggest that there is no reason to believe that this signal comes from anywhere other than an EL2 defect in GaAs. We also note that with the removal of the MOVPE EL2 data there is no longer the possibility of unintentionally comparing DLTS data from different device structures. The HVPE DLTS data are now presented on their own for the consideration of the reader.

Lastly, the carrier concentration was measured directly in every individual sample by capacitance-voltage, and these values were used in the calculation of trap densities, which we now note explicitly in the manuscript.

Additionally:

- Growth rate discussion – it would be helpful to add absolute values for the efficiency of Ga and AsH₃ utilization in these experiments. This is essential for the cost of the overall process.

We have estimated up to 70% Ga utilization in our research reactor over small areas, but we feel that these numbers could be misleading to report without a more rigorous study.

- I keep my opinion that the discussion of cap layer doping control should be removed and is a minor technical detail, not suitable for a Nature Communication article

We've moved these data to a supplementary figure.

In response to my earlier remark 1 and 2, I have been reading the two references of K. Schulte in App. Phys. Letters and K. Grüter in J. Cryst. Growth and agree that these papers discuss the influence of HVPE reactor pressure on growth rate sufficiently.

Thank you again to the reviewers for their time and effort to carefully review our manuscript. We believe our revised manuscript is clearer and stronger after addressing their concerns, and hope that it is now suitable for publication in Nature Communications.

Best Regards
Wondwosen Metaferia (Ph.D)
National Renewable Energy Laboratory

REVIEWERS' COMMENTS:

Reviewer #2 (Remarks to the Author):

I think the authors properly did a great job of taking the reviewer's comments and adding to the paper. I have no more suggestions and comments.

Reviewer #3 (Remarks to the Author):

The authors have successfully reorganized their manuscript and focused on the most important and clear findings. I believe the paper was substantially improved. I am happy with the new paper draft and would recommend it for publication after the following changes are made:

- Page 2 reference 8 does not demonstrate 200 mm HVPE growth but only 150 mm HVPE growth which is 6 inch diameter. Please correct.
- Page 13: the reference 16 is still not cited in an appropriate way. The reference shows an increase of the EL2 defect density by one order of magnitude only for 640°C but not for 680°C where the measured EL2 density was $<2E14$ cm⁻³ at 56 μ m/h and comparable to the results of the present paper. Therefore, one can only state that under non-ideal MOVPE conditions, a significant increase in EL2 density with increasing growth rate was observed in reference 16.
- Figure 6 – it has to be noted somewhere that the given Growth time does not include the transfer time for the wafers from one chamber to the next, nor any time required for heating/cooling of wafers. Therefore the given Growth times are purely for the absorber layers. Tunnel diodes and other thin layers may also require low growth rates and result in Growth times exceeding significantly 1 minute in total.
- Figure 6c please change GaInP to GaInP
- Page 16 replace "operation" by "efficiency" and same sentence you may want to clarify "60s to grow the thickest GaInP and GaAs absorber layers"

It is appropriate to add the sections on cap layer doping and DLTS measurement into the supplementary information.

Reviewer #3 (Remarks to the Author):

The authors have successfully reorganized their manuscript and focused on the most important and clear findings. I believe the paper was substantially improved. I am happy with the new paper draft and would recommend it for publication after the following changes are made:

- Page 2 reference 8 does not demonstrate 200 mm HVPE growth but only 150 mm HVPE growth which is 6 inch diameter. Please correct.

We corrected this in the manuscript. We have now removed reference 8 and the new reference #8 demonstrate the 8 inch HVPE growth.

- Page 13: the reference 16 is still not cited in an appropriate way. The reference shows an increase of the EL2 defect density by one order of magnitude only for 640°C but not for 680°C where the measured EL2 density was $<2E14 \text{ cm}^{-3}$ at 56 $\mu\text{m}/\text{h}$ and comparable to the results of the present paper. Therefore, one can only state that under non-ideal MOVPE conditions, a significant increase in EL2 density with increasing growth rate was observed in reference 16.

We understand that this part may create confusion to readers. We rephrased the sentence and cited the reference appropriately.

- Figure 6 – it has to be noted somewhere that the given Growth time does not include the transfer time for the wafers from one chamber to the next, nore any time required for heating/cooling of wafers. Therefore the given Growth times are purely for the absorber layers. Tunnel diodes and other thin layers may also require low growth rates and result in Growth times exceeding significantly 1 minute in total.

We explicitly added language about the required transfer times and showed that the increase in total process time would be 10-20 sec, depending on the structure. We continue to assume that all layers, including thin

layers, use the same growth rate. We believe that it is likely that a future production system can be designed for this purpose.

- Figure 6c please change GainP to GaInP

We thank the reviewer for catching this, we now corrected this typo.

- Page 16 replace “operation” by “efficiency” and same sentence you may want to clarify “60s to grow the thickest GaInP and GaAs absorber layers”

It is appropriate to add the sections on cap layer doping and DLTS measurement into the supplementary information.

Thank you again to the reviewers for their time and effort to carefully review our manuscript. We believe our revised manuscript is clearer and stronger after addressing their concerns, and hope that it is now suitable for publication in Nature Communications.

Best Regards
Wondwosen Metaferia (Ph.D)
National Renewable Energy Laboratory